# Rheological transition driven by matrix makes cancer spheroids resilient under confinement

Tavishi Dutt[1],*, Jimpi Langthasa[2],* , Monica Umesh[2,3], Satyarthi Mishra[1], Siddharth Bothra[3], Kottpalli Vidhipriya[2], Annapurna Vadaparty[4] , Prosenjit Sen[1,5] , Ramray Bhat[2,5]

Cancer metastasis through confining peritoneal microenvironments is mediated by spheroids: clusters of disseminated cells. Ovarian cancer spheroids are frequently cavitated; such blastuloid morphologies possess an outer ECM coat. We investigated the effects of these spheroidal morphological traits on their mechanical integrity. Atomic force microscopy showed blastuloids were elastic compared with their prefiguring lumenless moruloid counterparts. Moruloids flowed through microfluidic setups mimicking peritoneal confinement, exhibited asymmetric cell flows during entry, were frequently disintegrated, and showed an incomplete and slow shape recovery upon exit. In contrast, blastuloids exhibited size-uncorrelated transit kinetics, rapid and efficient shape recovery upon exit, symmetric cell flows, and lesser disintegration. Blastuloid ECM debridement phenocopied moruloid traits including lumen loss and greater disintegration. Multiscale computer simulations predicted that higher intercellular adhesion and dynamical lumen make blastuloids resilient. Blastuloids showed higher E-cadherin expression, and their ECM removal decreased membrane E-cadherin localization. E-cadherin knockdown also decreased lumen formation and increased spheroid disintegration. Thus, the spheroidal ECM drives its transition from a labile viscoplastic to a resilient elastic phenotype, facilitating their survival within spatially constrained peritoneal flows.

## Introduction

The peritoneal cavity of a patient suffering from advanced epithelial ovarian cancer is filled with disseminated multicellular aggregates, commonly known as spheroids (1). These spheroids colonize abdominal organs leading to metastasis (2). Metastasizing spheroidal cancer cells frequently become resistant to chemotherapeutic drugs necessitating a rigorous investigation of mechanisms underlying their formation and stability. Spheroids obtained by tapping the malignant ascites of ovarian cancer patients show heterogeneous morphologies: some exhibit a dysmorphic "moruloid" (mulberry-like) phenotype, and others show smooth compacted surfaces and an internal lumen, giving them a "blastuloid" appearance (3, 4). On the one hand, these diverse multicellular phenotypes could be aggregative consequences of phenotypically heterogeneous cell types that are shedded into the peritoneum. On the other hand, they could represent progressive stages of metastasis with moruloid phenotypes maturing into their blastuloid counterparts (5).

There is a burgeoning body of literature on biophysical investigations of tumorigenic cellular ensembles. Of these, most studies focus on the migrational dynamics of spheroid or tumoroid cells within stromal-like ECM microenvironments (6, 7, 8). The dynamics of ECM confinement, local growth, and mesoscale processes such as adhesion and diffusive gradients of signaling and proteolysis are relevant to such contexts. However, in fluid microenvironments, the assembly of multicellular structures from suspended single cells likely employs distinct mechanisms (9, 10). Although elegant theoretical models have been constructed recently to explain dynamical structural transitions, technical difficulties of efficiently imaging floating clusters have allowed few biophysical characterizations of spheroids (3, 11). Notable experimental exceptions include efforts to mechanically analyze spheroids using microtweezers, wherein those constituted from breast cancer cells were found to be softer than from their untransformed controls (12), and investigations using cavitational rheology to determine the cortical tension in spheroids of HEK293 cells (13). A pertinent study by Panwhar and coworkers recently describes a high-throughput approach using virtual liquid-bound channels to show that the stiffness of multicellular spheroids is an order of magnitude lower than that of cells that constitute them (14). Although these investigations have not studied temporal topological transitions between multicellular morphologies, they lay the foundation for such studies within fluid microenvironments.

[1]Centre for Nanoscience and Engineering, Indian Institute of Science, Bangalore, India  [2]Department of Developmental Biology and Genetics, Indian Institute of Science, Bangalore, India  [3]Undergraduate Program, Indian Institute of Science, Bangalore, India  [4]Shankara Cancer Hospital and Research Centre, India  [5]Department of Bioengineering, Indian Institute of Science, Bangalore, India

Correspondence: prosenjits@iisc.ac.in; ramray@iisc.ac.in
*Tavishi Dutt and Jimpi Langthasa contributed equally to this work

Whereas detachment from adhesive substrata generates cellular stresses (15), additional stresses are imposed on clusters because of movement through fluid spaces (16, 17, 18). Circulatory spaces such as the peritoneum have confining tunnels (19) and micro-cavities such as stomata (20) that cancer spheroids must negotiate safely for their survival and eventual metastasis.

In this study, we have probed the mechanical properties of ovarian cancer spheroids upon their traversal through a spatially constraining microfluidic channel. Previous studies have validated microfluidic approaches to be particularly useful in analyzing the mechanical properties of biomaterials, such as localized forces like traction, and their recovery dynamics (21, 22). We demonstrate that moruloid and blastuloid spheroids behave distinctly under constrictive flow. Our biophysical investigations in combination with multiscale modeling reveal the importance of the spheroidal ECM in multicellular morphological transitions and provide insight into the sustained endurance of spheroids within spatially confined geometries of the peritoneal cavity.

# Results

## A cavitational blastuloid phenotype shows elastic behavior

Moruloid (lumenless) and blastuloid (lumen-containing) spheroids form in a temporally successive progression from the ovarian cancer cell line OVCAR-3 upon suspended cultivation in low adherent conditions (Fig 1A; red and white represent fluorescent signals for F-actin and DNA, respectively). We first probed for differences in their bulk mechanical behavior using atomic force microscopy (AFM). To stabilize spheroids for AFM, we suspended them on top of agar beds (schematic shown in Fig 1Bi; see the Materials and Methods section for a detailed description of the agar preparation). We observed that blastuloids showed a higher mean elastic modulus compared with moruloids (Fig 1Bii; $P$ = 0.01; see Supplemental Data 1 for statistics). To further probe their mechanical properties under suspension, we subjected them to flow assays within a polydimethylsiloxane (PDMS)-constructed microfluidic channel that was constructed to simulate confining fluid microenvironments in the peritoneal cavity and that would constrain their movement (deformation spheroidometry) (Fig 1Ci). High-speed videography showed that moruloids emerged from the constrictive channel by assuming a shape that was deformed relative to their entry shape (Fig 1Cii; Video 1). We further quantified the deformation by measuring the aspect ratio of the leading edge of the spheroids as they exited the microfluidic channel. The aspect ratio was defined as follows:

$$Aspect\ Ratio\ (AR) = \frac{L}{w},$$

where $w$ is the minor axis, and $L$ is the major axis length of the protruding front end of the spheroids (Fig S1). These were quantified using a curve-fitting equation representing that of an ellipse (see the Materials and Methods section for further details). The parameters returned from the fit were used to obtain the aspect ratio for each individual protruded spheroid, for different time

points during its exit. A change in the aspect ratios was tracked as a function of time for representative clusters of each subset (Fig S1). The slopes were obtained by employing a simple linear fit on this curve. In contrast to moruloids, blastuloids were observed to regain their shape upon exiting the channel (Fig 1Ciii; Video 2). Consistent with these observations, the mean aspect ratio slope for moruloids was higher than blastuloids indicating the minor axis for the former lengthened in proportion to the major axis as the spheroids progressively moved out of the channel (Fig 1Civ; $P$ < 0.0001; see Supplemental Data 1 for statistics). The difference in ingress and egress characteristics was also observed for moruloids and blastuloids derived from G1M2 ovarian cancer patient xenograft cells (Fig S2; see also Video 3 and Video 4) and from freshly harvested spheroids of a high-grade serous ovarian cancer patient (Fig S2; see also Video 5 and Video 6). We next traced the change in the minor axis of spheroids normalized to their pre-entry values after spheroids had exited the channel to study the kinetics of their relaxation post-deformation. Moruloids showed variable but incomplete extents of recovery (Fig 1Di); in contrast, blastuloids relaxed very quickly (Fig 1Dii) with the mean normalized minor axis ratio (normalized to pre-ingress shape) that was higher than for moruloids (Fig 1Diii; $P$ = 0.04; see Supplemental Data 1 for statistics) and a mean minor axis peak time period (time taken for the minor axis to plateau) that was lower than for moruloids (Fig 1Div; $P$ = 0.07; see Supplemental Data 1 for statistics).

We next asked whether the kinetics of entry and transit of such spheroids illuminate the differences in their mechanical behavior observed above. To verify how the motility of the spheroids through constrictive spaces would depend on their morphological parameters, we measured entry time: defined as the time taken by the spheroid to fully enter the channel and calculated as the difference between the times when the leading edge first touches the channel, and the rear edge of the cluster first touches the channel (Fig 1Ei).

The entry time for individual moruloids and blastuloids along with their area (in pixels) was measured. We observed a moderate correlation of the temporal metrics with moruloid size ($R^2$ = 0.4) indicating bigger spheroids took longer to travel through the channel (Fig 1Eii). In contrast, the entry times for blastuloids were poorly correlated with spheroid size ($R^2$ = 0.01), indicating the mechanism by which blastuloids navigated the channel were relatively independent of their size (Fig 1Eiii). We next asked whether the increased time for entry observed for moruloids was due to rearrangements in intercellular contacts because of travel through constrictive spaces.

## Intercellular positional variation occurs within moruloids during entry into channel

To comprehend the flow of cells within spheroids when they traverse the channel, we used the PIVlab plugin in MATLAB to overlay morphologies with vectors (yellow) (23, 24). In order to measure the displacements of spheroidal contents during flow and their associations within different areas of the spheroids, we measured the mean flow angle within four regions of interest (at the top, bottom, front, and back) of moruloids and blastuloids (taking care to avoid the lumen of the latter) (Fig 2Ai and Aii). Angles were normalized to the direction of the flow of the spheroid, and the mean flow angle

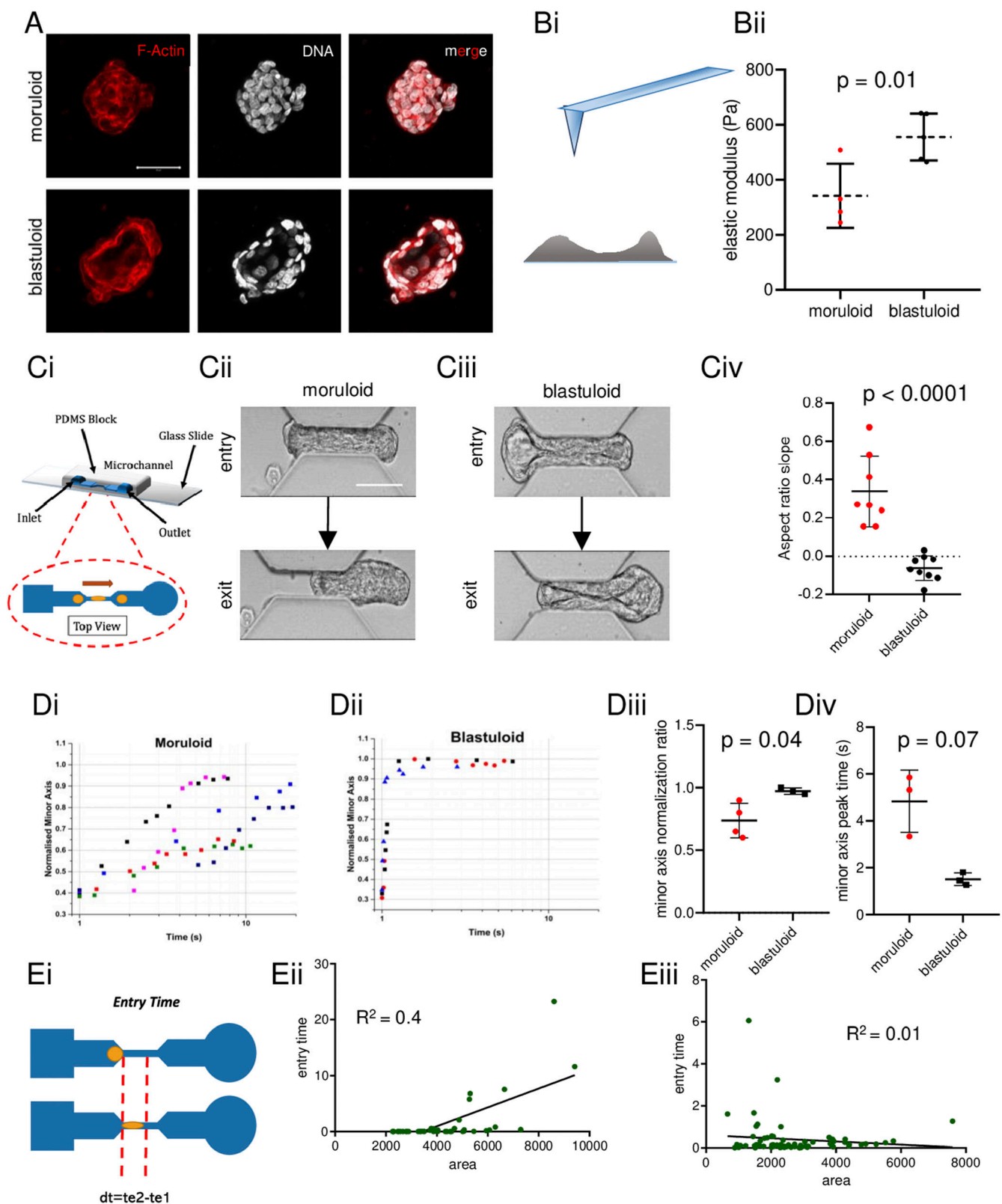

**Figure 1. Ovarian cancer blastuloids relax efficiently and rapidly upon deformation.**
**(A)** Laser confocal micrographs of moruloid (top) and blastuloid (bottom) OVCAR-3 spheroids showing maximum intensity projections of the fluorescence values representing F-actin (phalloidin; red) and DNA (DAPI; white). **(Bi)** Schematic depiction of atomic force microscopy performed on spheroids placed on agar beds. **(Bii)** Graph showing elastic moduli of moruloids (red dots) and blastuloids (black dots) (error bars, mean ± SD). **(Ci)** Schematic depiction of a microfluidic channel chip used for

was calculated for the four ROIs within multiple spheroids. Both the mean and the standard deviations for moruloid ROIs were higher than for blastuloids (Fig 2Aiii; $P$ = 0.04; see Supplemental Data 1 for statistics). This suggested a relatively less correlated and heterogeneous flow for cells within moruloids along the direction of the spheroidal flow compared with blastuloids.

As moruloids exited, the vectors overlaid on egressed cells were still aligned to each other; this is consistent with the deformation and retarded relaxation seen for moruloids in our earlier experiments (Fig 2Bi; see also Fig S2). During exit, vectors overlaid on egressing blastuloid cells diverged away from each other indicating the rapid relaxation and deformation recovery observed before (Figs 2Bii and S2).

This led us to ask whether the lack of flow correlation seen during entry of moruloids makes them vulnerable to cell detachment and spheroidal fractures. Upon close examination of the videomicrography of moruloids, we observed evidence of cell detachment and spheroidal damage and disintegration (Fig 2C; see also Fig S3 for a high-speed per-second camera observation of spheroidal cell detachment). Extending our observations across a wide number of traversals, a higher proportion of moruloids were found to encounter damage compared with blastuloids (Fig 2Di; $P$ = 0.04; see Supplemental Data 1 for statistics). Moreover, the mean size of moruloids beyond which disintegration was observed was lower than that of blastuloids (Fig 2Dii; $P$ = 0.02; see Supplemental Data 1 for statistics). These observations suggest that both spheroidal size and phenotype regulate morphological integrity in constrained movement. Our observations of intercellular detachment as an output of both these traits led us to investigate the effects of cell–cell adhesion on the constrictive traversal of distinct morphologies.

### A multiscale model predicts lower cell adhesion in moruloids

Multiscale computational models have been used with a great deal of success to simulate experimentally elucidated observations and verify whether multicellular phenotypes can indeed emerge through the empirical interactions (25, 26, 27). CompuCell3D represents one of the best-known simulation frameworks allowing an integration of cellular Potts model–based solving of a Hamiltonian for contact energies between cellular constituents with PDE solvers of molecular constituents (28).

We therefore constructed multicellular models for the moruloids and blastuloids, which were driven through a constrictive channel in the simulation space, similar to our experimental setup. The input parameters that were tunable within such simulations were the contact energies (inverse of adhesion strength) between individual digital cells. We performed parameter scans for a range of intercellular contact energy values (refer to Table 2). We observed

stable "digital moruloids" formed for all the 4 (0–15) values of contact energies tested. For stable "digital blastuloid" formation, the contact energy values between "core cells" (which represent lumen), and those between the core cells and the peripheral cells (which represent the digital equivalents of biological cells) were fixed at 0 (high adhesion), whereas the contact energy values between peripheral cells were varied from 5 to 15 (see Table 2). For these stable spheroid conditions, we confirmed that upon exiting the simulated constrictive channel space, "digital moruloids" showed a sustained deformation and intercellular rearrangement (through observed intermixing of peripheral [blue] and central [green] cells of the spheroids) (Fig 3A and Video 7). On the other hand, the digital blastuloids recovered their shape rapidly upon exit (Fig 3B and Video 8). Insets of exiting digital moruloids and blastuloids showed intercellular rearrangements in the former but not the latter, showing consistency with our experimental findings (Fig 3A and B, insets to the right). Consistent with these observations, the aspect ratio of egressing digital moruloid morphologies when normalized to the entry aspect ratio was found to be significantly higher than that for digital blastuloid counterparts (unpaired $t$ test, $P$ < 0.0001; Fig 3C). In fact, at lower values of intercellular adhesion (contact energy = 15), we also observed cell detachment and disintegration for both digital moruloids (Fig 3D, Video 9) and blastuloids (Fig 3E, Video 10). In agreement with the experimental observations, we found greater cell detachment in the case of digital moruloids (46.67%) than with the digital blastuloids (20%) (Fig 3F). Upon increasing intercellular adhesion, a decrease in the normalized aspect ratio was observed for digital moruloids, suggesting that cell–cell adhesion may regulate the unjamming-to-jamming transition of multicellular ensembles (one-way ANOVA, $P$ < 0.0001; Fig 3G).

Interepithelial adhesion is principally mediated through basolateral adherens junctions, established using homodimeric interactions between transmembrane cell adhesion molecules such as E-cadherin. The expression of E-cadherin has been shown to be under the regulation of Laminin 111, a principal constituent of epithelial basement membrane (BM) matrix: the latter decreases DNMT1 levels in breast cells resulting in a decrease in promoter methylation of the *CDH1* gene that encodes for E-cadherin (29). In fact, work across several systems has established an instructive role of BM matrix in establishing the apicobasal polarity as is evidenced in blastuloids (e.g., in MDCK cysts (30), kidney and lung organotypic cultures (31, 32), and murine embryoid bodies (33)). We have demonstrated in an earlier study that an increased expression and relocalization of BM-typical proteins such as Fibulin, Collagen IV, and Laminins are associated with blastuloid formation (3). Our observation that blastuloids showed lesser cell detachment and our theoretical prediction of an association of higher intercellular

performing the flow experiments. **(Cii)** Snapshots of high-speed time-lapse videography moruloids at the entry (top) and exit (bottom) of the channel (see also Video 1). **(Ciii)** Snapshots of high-speed time-lapse videography blastuloids at the entry (top) and exit (bottom) of the channel (see also Video 2). **(Civ)** Graph showing aspect ratio slopes of exiting moruloids and blastuloids (see also Fig S1) (error bars, mean ± SD). **(Di, Dii)** Representative traces showing a change in the minor axis of exited relaxing moruloids (Di) and blastuloids (Dii). **(Diii)** Graph depicting minor axis normalization ratios for exited relaxing moruloids and blastuloids (error bars, mean ± SD). **(Div)** Graph depicting minor axis peak time of exited relaxing moruloids and blastuloids (error bars, mean ± SD). **(Ei)** Schematic representation of entry time. **(Eii)** Time–size correlation plots of moruloids for entry time. **(Eiii)** Time–size correlation plots of blastuloids for entry time. Significance was computed using Tukey's multiple comparisons test (see Supplemental Data 1 for statistics). Scale bars = 50 $\mu m$. Each result is derived from ≥3 independently performed experiments.

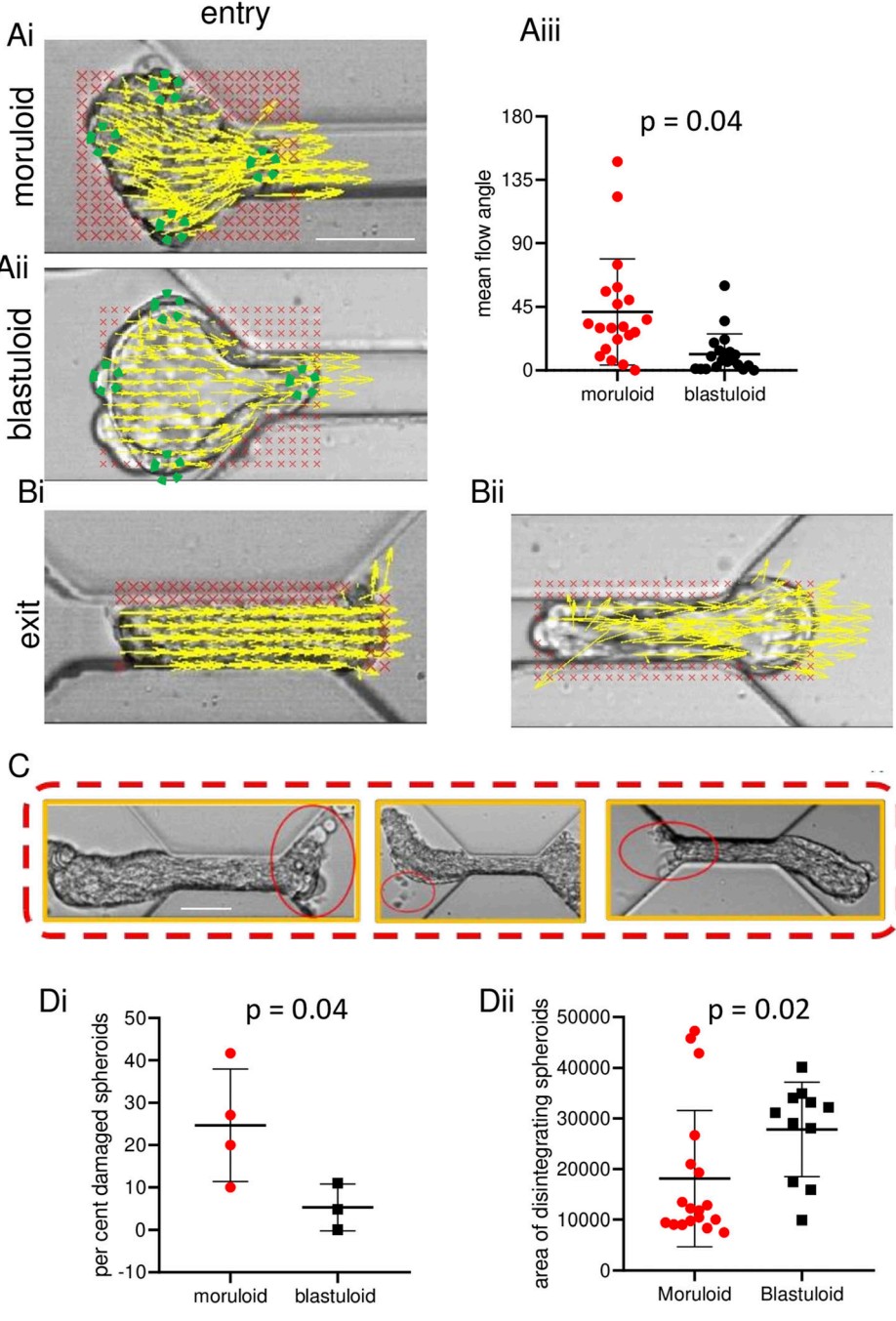

**Figure 2. Ovarian cancer blastuloids exhibit minimal intercellular rearrangement.**
**(Ai, Aii)** Particle image velocimetry shows velocity vectors sized based on magnitudes (yellow) overlaid on an image of ingressing moruloid (Ai) and blastuloid (Aii). **(Aiii)** Graph showing mean flow angles of ingressing moruloids and blastuloids (error bars, mean ± SD). **(Bi, Bii)** Particle image velocimetry shows vectors sized based on magnitudes (yellow) overlaid on an image of ingressing moruloid (Bi) and blastuloid (Bii). **(C)** Representative images showing cell detachment and disintegration in moruloid spheroids as they traverse the constrictive channel; red ellipses highlight damage (see also Fig S3). **(Di, Dii)** Graph showing percent spheroid damage of exiting moruloids and blastuloids (Di) (error bars, mean ± SEM) and areas of damaged spheroids for moruloids and blastuloids (Dii) (error bars, mean ± SD). Significance was computed using Tukey's multiple comparisons test (see Supplemental Data 1 for statistics). Scale bars = 50 $\mu$m. Each result is derived from ≥3 independently performed experiments.

adhesion with a lower aspect ratio led us to ask whether the blastuloids expressed comparatively greater levels of E-cadherin. Using fluorescent immunocytochemistry, we observed greater transmembrane signals for E-cadherin in blastuloids compared with moruloids (Fig 3H; green depicts signals for E-cadherin; red and white denote signals for F-actin using phalloidin and DNA using DAPI). We have shown before that blastuloid architectures (but not moruloids) show cavitation and an ECM coat (3). When treated with collagenase, most of the blastuloids (75–90%) lose the cavitation. On the other hand, when cultured in suspension in the presence of a 2% laminin-rich basement membrane, moruloids develop microlumina and central cavities faster than when in the absence of the matrix (Fig S4). Upon ECM degradation, cavitation is lost (see Fig S5). In addition, the intercellular staining of E-cadherin in collagenase-treated blastuloids was found to be weaker and more diffuse from lateral junctional boundaries than control untreated blastuloids (Fig 3I).

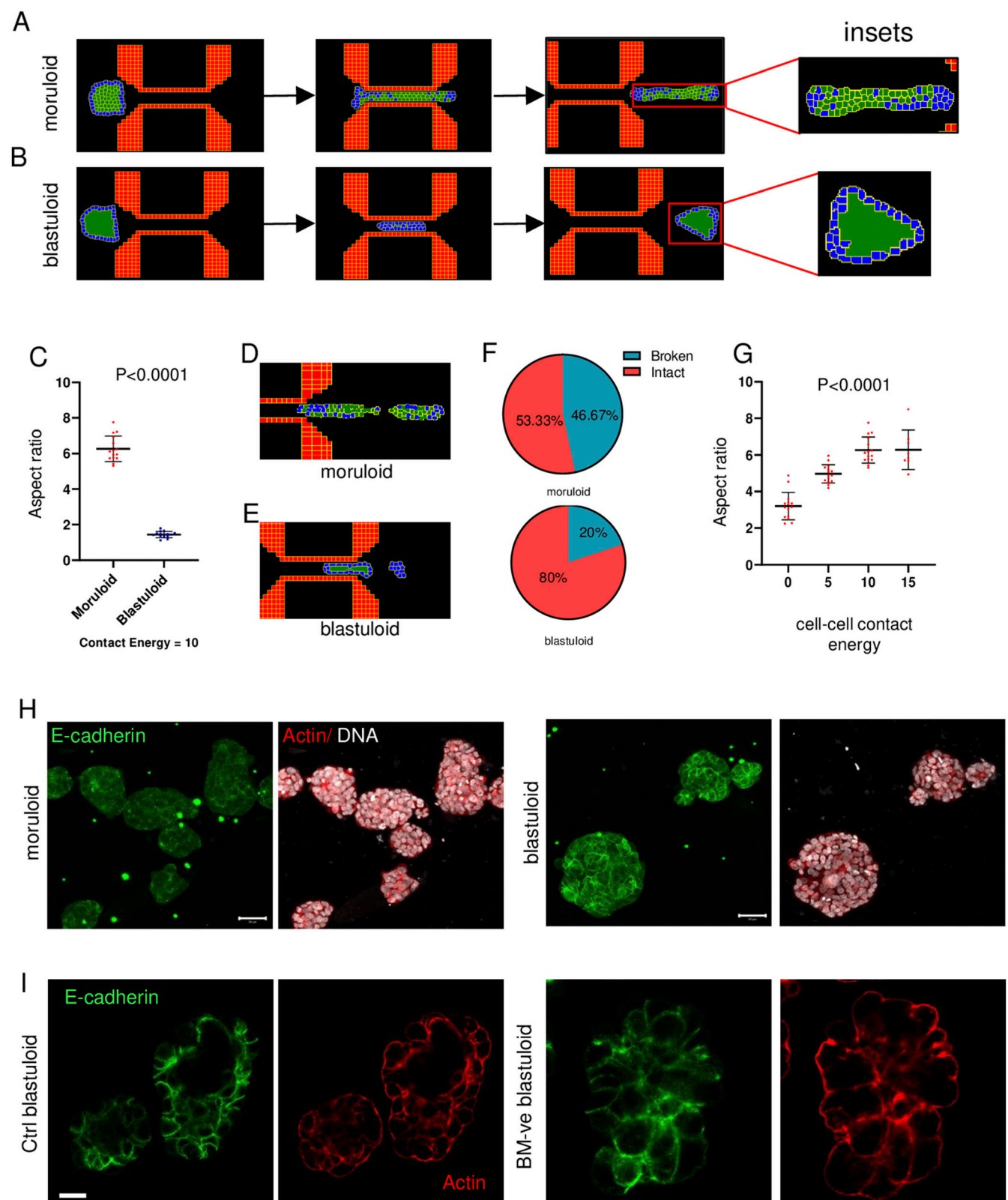

**Figure 3. Ovarian cancer blastuloids show higher intercellular adhesion.**
**(A, B)** Snapshots of CompuCell3D simulations of digital moruloid (A) and blastuloid (B) spheroids traversing through a spatially constrictive channel representing time points when the spheroids enter the channel (left), inside the channel (middle), and during exit from the channel (right). Insets show the intercellular arrangement within the digital spheroids upon exit. **(C)** Scatter graph showing the difference between aspect ratios of digital moruloid and blastuloid spheroids upon exit from the channel

## ECM regulates the mechanical behavior of blastuloids

We next asked whether blastuloid ECM distinguishes its mechanics from moruloids. We had earlier shown that an enzymatic debridement of BM ECM leads to loss of lumen and washing away the lumen restores the blastuloid phenotype (3). Blastuloids were enzymatically debrided of their ECM coat using type IV collagenase (Fig S6; green and white represent fluorescent signals for type IV collagen and DNA, respectively). AFM showed that the mean elastic modulus of ECM-removed blastuloids was significantly lower than blastuloids and closer to moruloids (one-way ANOVA, $P = 0.008$; see Supplemental Data 1 for statistics; Fig 4A). A time-lapse videographic (Video 11) examination showed that ECM-removed blastuloids deform as they exited the microfluidic channel like moruloids (Fig 4B) and their mean aspect ratio slope was similar to moruloids and distinct from blastuloids (one-way ANOVA, $P < 0.0001$; see Supplemental Data 1 for statistics; Fig 4C). The relaxation kinetics of exited ECM-removed blastuloids was found to recover only partially (Fig 4Di–iii); minor axis normalization ratio (Fig 4Dii) and minor axis peak time were similar to moruloids (Fig 4Diii) (one-way ANOVA, $P = 0.01$ and $0.02$, respectively, for the two metrics; see Supplemental Data 1 for statistics). The entry time of ECM-removed blastuloids showed moderate correlation with spheroidal size, resembling moruloid rather than blastuloid traits ($R^2 = 0.22$; Fig 4E). Particle image velocimetry (PIV) on ECM-removed blastuloids entering the channel revealed heterogeneous and large mean flow angle similar to moruloids (Fig 4F; one-way ANOVA, $P = 0.01$; see Supplemental Data 1 for statistics). ECM removal in blastuloids also led to an increased damage compared with blastuloids (Fig 4Gi; one-way ANOVA, $P = 0.03$; see Supplemental Data 1 for statistics) with the maximum size at which damage was observed being lower than that for blastuloids (Fig 4Gii; one-way ANOVA, $P < 0.001$; see Supplemental Data 1 for statistics).

Because we had observed that a loss in ECM led to a decrease in E-cadherin membrane localization, we investigated whether E-cadherin depletion in spheroids grown to blastuloid stage phenocopied ECM debridement in the latter's effect on spheroid mechanics. We observed a decrease in lumen formation in blastuloids constituted from OVCAR-3 shCDH1 cells compared with scrambled shRNA controls (Fig 5A; see Fig S7 for demonstration of E-cadherin KD upon stable lentiviral transduction of cognate shRNA). High-speed time-lapse videography of their movement through our microfluidic channel showed an exit, where an immediate relaxation was not evident (Fig 5B): this was reflected in the aspect ratio slope whose mean magnitude was significantly higher than control blastuloids (Fig 5C; unpaired $t$ test, $P = 0.001$; see Supplemental Data 1 for statistics). The mean entry time for shCDH1

OVCAR-3 moruloid showed a moderate correlation with their sizes ($R^2 = 0.22$; Fig 5D; see also Fig S8 for correlation measurements of mean entry time and size of scrambled control OVCAR-3 blastuloids). The mean flow angle of ROIs within the spheroids compared with control blastuloids suggested cell flows within the spheroids were poorly correlated with the overall movement (Fig 5E; unpaired $t$ test, $P = 0.01$; see Supplemental Data 1 for statistics). This led us to closely observe the traversal of these spheroids through the channel and calculate the percentage that showed signs of damage and fractures: the mean proportion of damaged spheroids and the mean size at which spheroid damage occurred were respectively higher and lower than those for blastuloids (Fig 5Fi and Fii; unpaired $t$ test, $P = 0.04$ and $0.07$, respectively; see Supplemental Data 1 for statistics). These quantitative observations suggested that spheroids made with E-cadherin–depleted cells were less resilient than blastuloids.

We asked whether overexpressing E-cadherin in OVCAR-3 cells would allow their constituted moruloids to mimic blastuloids in resilience (Fig S7). We observed insignificant differences in the traversal kinetics of E-cadherin–overexpressing and empty vector–expressing control moruloids in terms of spheroidal morphology at entry and exit (Fig 5G). Their mean aspect ratio slopes were also insignificantly altered (Fig 5H; unpaired $t$ test, $P = 0.8$; see Supplemental Data 1 for statistics). E-cadherin–overexpressing moruloids showed a mild correlation of their entry times like their controls (Fig 5I; $R^2 = 0.03$; see also Fig S8). Their mean flow angles (Fig 5J; unpaired $t$ test, $P = 0.11$; see Supplemental Data 1 for statistics), proportion of damage (Fig 5Ki; unpaired $t$ test, $P = 0.82$; see Supplemental Data 1 for statistics), and mean area at which damage of spheroids is observed (Fig 5Kii; unpaired $t$ test, $P = 0.18$; see Supplemental Data 1 for statistics) were unaffected by E-cadherin overexpression. This suggested that the mere overexpression of E-cadherin was unlikely to alter the mechanics of moruloid morphologies. The presence of a lumen along with robust E-cadherin expression as driven by the ECM coat was responsible for resilience under dynamical constraint.

## Discussion

Our study is centered on an important observation: that the same cancer cell can form distinct multicellular architectures with unique mechanical properties. A spheroid that is constituted entirely of cells with no internal lumen exhibits deformability upon strain because of spatially constrictive travel. In addition, their shape recovery timescales are far slower than their deformation

normalized to the same at entry (error bars, mean ± SD). **(D, E)** Snapshot of a CompuCell3D simulation of a digital moruloid (D) and blastuloid (E) spheroid showing disintegration and cell detachment upon traversal. **(F)** Pie chart representing the fraction of spheroid disintegration upon traversal for digital moruloid (top) and blastuloid (bottom) spheroids. **(G)** Scatter graph of the normalized aspect ratios of digital moruloid spheroids with different values of cell–cell contact energy used in the multiscale simulation. One-way ANOVA was used to compute statistical significance (error bars, mean ± SD). **(H)** Laser confocal micrographs of moruloid and blastuloid OVCAR-3 spheroids showing maximum intensity projection of the fluorescence values representing E-cadherin (green) and counterstaining for F-actin (phalloidin; red) and DNA (DAPI; white). **(I)** Laser confocal micrographs of blastuloid OVCAR-3 spheroids (untreated control left) and upon type IV collagenase treatment (right) showing middle stack of the fluorescence values representing E-cadherin (red) and counterstaining for F-actin (phalloidin; green) (see also Fig S5). **(C, G)** Significance was computed using an unpaired $t$ test for (C) and Tukey's multiple comparisons test for (G) (see Supplemental Data 1 for statistics). Scale bars = 50 $\mu$m. Each result is derived from ≥3 independently performed experiments.

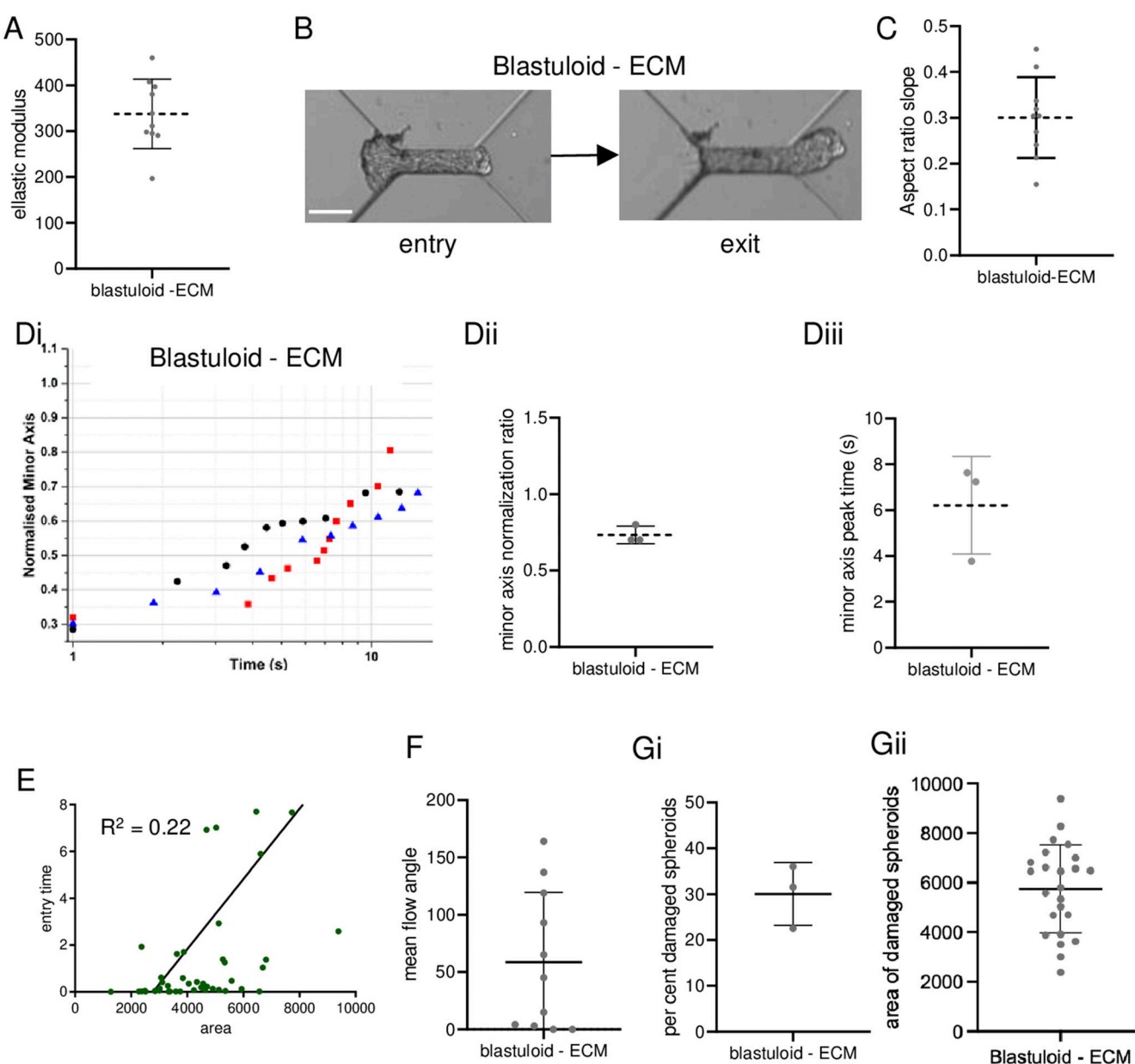

**Figure 4. ECM removal in blastuloids results in mechanical behavior typical of moruloids.**
**(A)** Graph showing elastic moduli of blastuloids upon ECM removal (error bars, mean ± SD). **(B)** Snapshots of high-speed time-lapse videography blastuloids with ECM removal at the entry (left) and exit (right) of the channel (see also Video 11). **(C)** Graph showing the aspect ratio slopes of exiting blastuloids upon ECM removal (see also Fig S1) (error bars, mean ± SD). **(Di)** Representative traces showing a change in the minor axis of exited relaxing ECM-removed blastuloids. **(Dii)** Graph showing the minor axis normalization ratio for exited relaxing ECM-removed blastuloids (error bars, mean ± SD). **(Diii)** Graph showing minor axis peak time of exited relaxing ECM-removed blastuloids (error bars, mean ± SD). **(E)** Graphs showing size–time correlation plots of ECM-removed blastuloids for entry time. **(F)** Graph showing mean flow angles for ingressing ECM-removed blastuloids (error bars, mean ± SD). **(Gi, Gii)** Graph showing percent spheroid damage of exiting ECM-removed blastuloids (Gi) (error bars, mean ± SEM) and areas of damaged ECM-removed blastuloids (Gii) (error bars, mean ± SD). Significance was computed using Tukey's multiple comparisons test (see Supplemental Data 1 for statistics). Scale bars = 50 $\mu$m. Each result is derived from ≥3 independently performed experiments.

timescales indicating irreversible changes in their mechanical structure indicative of plastic behavior. Under strain, they also show damage. On the other hand, a lumen-containing spheroid shows a jammed cellular organization with a low propensity for cellular detachment. Post-deformation, their recovery timescales are much faster and complete, suggesting their morphology undergoes little change under construction. Although this could be indicative of an elastic behavior of materials, their shorter entry times suggest that blastuloids are able to maintain their morphology by contextually reducing their luminal volume during entry. In fact, initial attempts to computationally model the blastuloids as impermeable elastic spheres resulted in longer entry times in simulations as well. This was mitigated upon simulating a dynamic deflation–inflation behavior of the digital blastuloids during traversal. The kinetics of

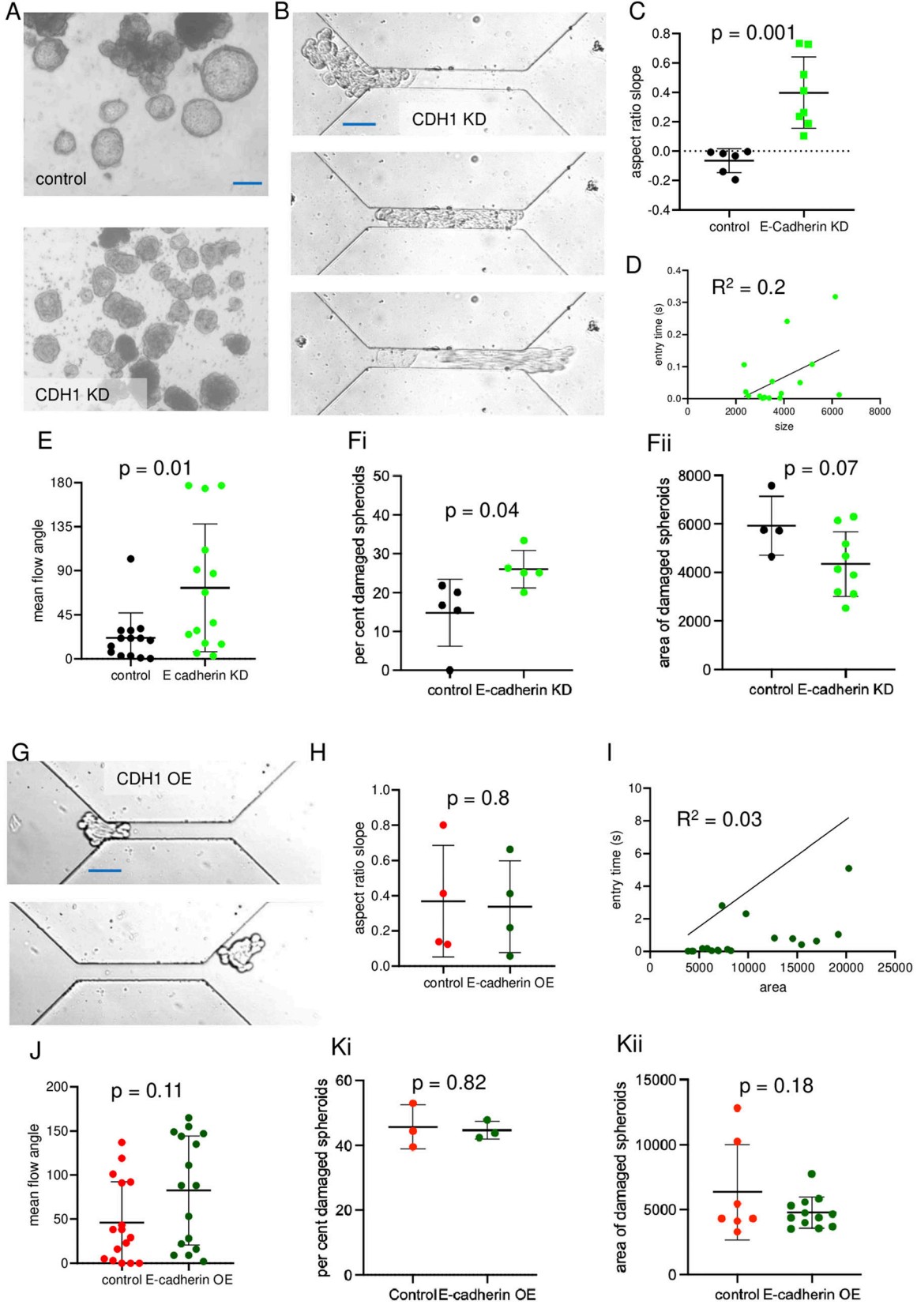

lumen reinflation of these spheroids upon exiting the constriction suggests that the lumen acts as a dynamical volume whose magnitude is a function of the strain faced by the spheroid. Furthermore, the similarity between the behavior of blastuloids upon ECM removal, and moruloids can be understood by how the basement membrane (BM) matrix mediates two crucial morphogenetic characteristics of blastuloids: maintenance of lumen and high intercellular adhesion. The lumenless spheroids on the other hand are morphogenetically labile, owing to a loss of the external BM and lower intercellular adhesion. Therefore, the kinetics of regaining a more spheroidal shape occurs on a longer scale in moruloids than for blastuloids. Furthermore, the recovery is incomplete as is indicated by our PIV analysis, because of an irreversibly altered arrangement of the cells constituting them.

As moruloids differentiate into blastuloids, the BM induces a transition in cell migratory dynamics from an unjammed to a jammed state (34, 35). It has not escaped our attention that the blastuloid-to-moruloid transition is also accompanied by a switch in cell state that could be likened to one resembling a mesenchymal state to an epithelioid counterpart (36, 37). We had explored aspects of this switch in our earlier study, where we showed that moruloid phenotypes were associated with intraspheroidal cell motility, a heightened expression of fibronectin, depleted levels of the BM matrix and the apical junctional protein ZO-1 (38). On the other hand, the blastuloid phenotype shows cell sessility, increased ZO-1 and BM proteins, and depleted fibronectin. In addition, the cells comprising blastuloids have a tetragonal polar shape as opposed to the heterogeneous moruloid morphologies. These data are further strengthened by our demonstration in the curr study of higher E-cadherin in blastuloid cells compared with moruloid counterparts (39). In addition, whereas a tendency to detach from a multicellular mass strongly characterizes a mesenchymal state, higher intercellular adhesion is typical of epithelial one (40). Fredberg and colleagues have explored whether the transition from unjammed to jammed cell behavior and that from mesenchymal to epithelial states are conceptually non-congruent across biological examples (40). Our study seeks to bridge the two frameworks by showcasing how a limited set of proteins shift multicellular states in terms of both their polarity and motility. A more rigorous annotation of moruloid and blastuloid cell states as being mesenchymal and epithelioid would require a systematic assessment of a larger set of established markers (41), which would be carried out in the future. The knockdown of E-cadherin performed in this study is stable and constitutive; a more ECM debridement–analogous perturbation would have been an induced knockdown in already formed

spheroids. However, a short temporal window for moruloid–blastuloid transition disallows such inducible perturbations to fully manifest before the establishment of the blastuloid phenotype. Nevertheless, we aim to perform such investigations in the future.

Do spheroids encounter spatially restricted spaces within the peritoneal cavity? Apart from the classical epiploic foramen that connects the lesser and greater sac, there are several peritoneal fossae and lymphatic stomata, wherein spheroids may potentially be subjected to constriction (42). Moreover, the peritoneal spaces are dynamic owing to constant motion within the surrounding organs, such as peristalsis. Such spatial restrictions could impinge on the rheology of diverse spheroidal morphologies and enhance the spectrum of their behaviors: cell detaching from the moruloids is better suited to micrometastatic colonization, whereas the jammed stable blastuloids could represent a niche suited for longer survival within the ascites. This is seen through the prolific accumulation of the blastuloids within the ascites of ovarian cancer patients.

Our study throws up engaging questions. On a mechanistic level, what is responsible for dynamical behavior of the blastuloids: the lumen, the BM, or the higher intercellular adhesion? The knockdown of the E-cadherin and the concomitant decrease in the spheroidal lumen (phenocopying for multiple traits of the moruloid and the ECM-removed blastuloids) indicate that these processes are entangled phenomenologically and that it is challenging to dissect the exact contribution of each of these to the ensemble properties. Genetic perturbation of matrix proteins in a temporally controlled manner will be undertaken in the future to dissect the origins of tissue elasticity. Multicellular ensembles have now been proposed to have greater potential for metastatic colonization than their unicellular counterparts. Could such potential be explained by their ability to withstand strain because of robustness to disintegration even in narrow peritoneal spaces? A broader generalization of our study to other cancers that employ both transperitoneal and vascular routes of dissemination would shed light on the influence of mechanobiological properties on disseminated cancer cell behavior.

# Materials and Methods

### Cell culture

Ovarian cancer cell lines used in this study are OVCAR-3 (American Type Culture Collection; kind gift from Prof Rajan Dighe, Indian

**Figure 5. E-cadherin depletion phenocopies ECM debridement.**
**(A)** Phase-contrast micrographs of scrambled control OVCAR-3 blastuloids and E-cadherin–depleted OVCAR-3 spheroids (see Fig S7, left) showing the presence of lumen in the former and absence in the latter. **(B)** Traversal of E-cadherin–depleted OVCAR-3 spheroids at ingress (top), within the channel (middle) and at exit (bottom). **(C)** Graph showing aspect ratio slopes of exiting scrambled control OVCAR-3 blastuloids and E-cadherin–depleted OVCAR-3 spheroids (error bars, mean ± SD). **(D)** Time–size correlation plots of control blastuloids for entry time (see Fig S8) and of E-cadherin–depleted OVCAR-3 spheroids for entry time. **(E)** Graph showing mean flow angles of exiting scrambled control OVCAR-3 blastuloids and E-cadherin–depleted OVCAR-3 spheroids (error bars, mean ± SD). **(Fi, Fii)** Graph showing percent spheroid damage of exiting scrambled control OVCAR-3 blastuloids and E-cadherin–depleted OVCAR-3 spheroids (Fi) (error bars, mean ± SEM) and areas of damaged spheroids (Fii) for exiting scrambled control OVCAR-3 blastuloids and E-cadherin–depleted OVCAR-3 spheroids (error bars, mean ± SD). **(G)** Traversal of E-cadherin–overexpressing OVCAR-3 spheroids (see Fig S7) at ingress (top) and at exit (bottom). **(H)** Graph showing aspect ratio slopes of exiting vector control OVCAR-3 moruloids and E-cadherin–overexpressing OVCAR-3 spheroids (error bars, mean ± SD). **(I)** Time–size correlation plots of control moruloids for entry time (see Fig S8) and of E-cadherin–overexpressing OVCAR-3 spheroids for entry time **(I)**. **(J)** Graph showing mean flow angles of exiting vector control OVCAR-3 moruloids and E-cadherin–overexpressing OVCAR-3 spheroids (error bars, mean ± SD). **(Ki, Kii)** Graph showing percent spheroid damage of exiting vector control OVCAR-3 moruloids and E-cadherin–overexpressing OVCAR-3 spheroids (Ki) (error bars, mean ± SEM) and areas of damaged spheroids (Kii) for exiting vector control OVCAR-3 moruloids and E-cadherin–overexpressing OVCAR-3 spheroids (error bars, mean ± SD). Significance was computed using unpaired $t$ test (see Supplemental Data 1 for statistics). Scale bars = 50 $\mu$m. Each result is derived from ≥3 independently performed experiments.

Institute of Science) and G1M2 (patient-derived xenograft line; gift from Prof Sharmila Bapat, National Centre for Cell Sciences, India). These cell lines were maintained in DMEM—high glucose (AL007A; HiMedia) and Roswell Park Memorial Institute medium (AL162A; HiMedia) supplemented with 10–20% FBS and recommended antibiotics in a humidified atmosphere of 95% air and 5% $CO_2$ at 37°C. Cell identities were confirmed through STR analysis and were routinely tested for Mycoplasma contamination.

### Spheroid culture and collagenase treatment

Spheroid preparation is detailed in reference 3. Briefly, 1.5–2 × 10⁵ cancer cells were cultured in suspension on low attachment plates (by coating them with the hydrogel polyHEMA) in a defined serum-free medium for 1 to 7 d. Moruloids with uncompacted berry-like appearance with an absence in lumen were observed within 24 h. Transition to a compacted state with multiple small lumen occurred for 75% of cultivated moruloids by 72 h post-cultivation, and by 7 d, 90% spheroids had switched to a blastuloid appearance with a central lumen and compacted surface. Medium composition was as follows: DMEM: F12 (1:1) (HiMedia AT140) supplemented with 0.5 $\mu$g/ml hydrocortisone (H0888; Sigma-Aldrich), 250 ng/ml insulin (I6634; Sigma-Aldrich), 2.6 ng/ml sodium selenite (S5261; Sigma-Aldrich), 27.3 pg/ml estradiol (E2758; Sigma-Aldrich), 5 $\mu$g/ml prolactin (L6520; Sigma-Aldrich), 10 $\mu$g/ml transferrin (T3309; Sigma-Aldrich). 1.5 × 10⁵ cells were seeded in 35-mm dishes for the experiments cultured with replenishment of medium at regular intervals. Spheroids were collected from the cultures by centrifugation. Moruloids were harvested at 24–48 h, and blastuloids were obtained after 7 d. Collagenolysis of blastuloids was performed by incubating with 600 U collagenase IV (17104-019; Gibco) for 24 h. Type IV collagenase is known to have the lowest non-collagenase–based trypsin-like proteolytic activity leaving cell membrane proteins intact. In fact, we performed appropriate experiments in our previous study (3), wherein the dose and the timing were optimized to give results that are specific to spheroidogenesis without impairing cell viability, to the extent that when the enzyme was washed away, the collapsed blastuloids regain structure in terms of lumen formation.

### Clinical samples

Ascites obtained from the peritoneal tap of patients with ovarian cancer was provided by Sri Shankara Cancer Hospital with due consent and ethical clearance from their Institutional Ethical Committee (SSCHRC/IEC4/015). Patient spheroids were cultured in tissue culture–treated polystyrene substrata/polyHEMA-coated dish using DMEM (AL007A; HiMedia)—supplemented with 10–20% FBS (10270; Gibco) and antibiotics or with defined medium. Spheroids were then collected from the cultures by centrifugation.

### Atomic force microscopy

A solution of noble agar dissolved in PBS was heated in a microwave oven. The solution was immersed partially in a water bath and heated until the agar was fully dissolved. Different concentrations of this solution were tried, resulting in optimal performance with 2% wt/vol (0.04 g of noble agar powder in 2 ml of PBS).

The molten solution was poured on the edge of a preheated (to 37°C) 35-mm petri dish, and a freshly cleaned glass slide is used to smear the dispensed molten agar. Cells or spheroids were dispensed using a micropipette as big drops (~100 $\mu$l) and allowed to rest for the agar to solidify and the cells to attach. 1 ml of media or Hepes buffer is put in the final step, and it is taken for the measurement. The AFM experiments were conducted using the Park Systems XE-Bio AFM tool. The parameters used for the measurements are described in Table 2. The software used for analysis is XEI software. The force–distance curves need to be converted to force–separation curves. However, this conversion is done by XEI software, which directly allows us to proceed from the force–separation curves.

Young's modulus values are obtained from the software that uses a Hertzian model to calculate the values, which is described by the following equation:

$$F = \frac{4}{3} \frac{E}{(1-\upsilon)^2} \sqrt{R\delta^3},$$

where
- F is the applied force.
- E is Young's modulus.
- $\upsilon$ is Poisson's ratio.
- R is the radius of the tip.
- $\delta$ is the indentation of the sample.

### Gene expression analysis

Cells were lysed in the manufacturer's recommended volume of TRIzol. RNA was isolated by the chloroform–isopropanol extraction method. All reagents were molecular grade and purchased from Merck. Quantification of RNA yield was performed using NanoDrop ND-1000 UV-Vis Spectrophotometer (NanoDrop Technologies). 1 $\mu$g of total RNA was reverse-transcribed using the Verso cDNA synthesis kit as per the manufacturer's protocol (AB-1453; Thermo Fisher Scientific). Real-time PCR was performed with 1:2 diluted cDNA using the SYBR Green detection system (F415L; Thermo Fisher Scientific) and Rotor-Gene Q (9001560; QIAGEN). The 18S rRNA gene was used as the internal control for normalization. Relative gene expression was calculated using the comparative Ct method, and gene expression was normalized to unsorted cells. The genes whose expressions are measured along with their primer sequences are mentioned in Table 1. Appropriate no template and no-RT controls were included in each experiment.

### Immunocytochemistry and imaging

Spheroids were collected after centrifugation at 200$g$ and fixed using 3.7% formaldehyde (24005; Thermo Fisher Scientific) at 4°C for 20 min. After one wash with PBS, the fixed spheroids were resuspended in PBS and 10–20 $\mu$l of spheroid suspension was put to an eight-well chambered cover glass followed by placing on a dry bath at 37°C for 15–30 min of drying. Spheroids were then permeabilized using 0.5% Triton X-100 (MB031; HiMedia) for 2 h at RT. Using 3% BSA

**Table 1. Table showing the parameters used for the atomic force microscopy measurements.**

| Nature of the probe | Spherical |
|---|---|
| Diameter of the probe | 5 µm |
| Cantilever | Hydra 0.05 nm |
| Force constant | $45 \times 10^{-3}$ (N/m) |
| Sensitivity | 35 (V/µm) |
| Forward speed | 0.8 (µm/s) |
| Backward speed | 0.8 (µm/s) |

(MB083; HiMedia) prepared in 0.1% Triton X-100 solution, blocking was achieved at RT and incubated for 45 min. Primary antibody against E-cadherin (24E10; Cell Signaling Technology) was incubated overnight at 4°C, which was followed by washes using 0.1% Triton X-100 in PBS (5 min × 3). Secondary antibody was incubated at RT for 2 h under dark conditions. DAPI (D1306; Thermo Fisher Scientific) was added to the samples and incubated for 15 min. Three washes were given after secondary antibody incubation step and after the addition of DAPI. Images were captured in 20X and 40X using a Carl Zeiss LSM 880 laser confocal microscope. Images were processed and analyzed using ZEN Lite software. Either for counterstaining or for non-immunocytochemical visualization, paraformaldehyde-fixed spheroids were washed with 1× PBS for 5 min thrice, blocked and permeabilized as above, and stained with 1 µg/ml DAPI and 1:500 Alexa Fluor 633/568–conjugated phalloidin (A22284; Thermo Fisher Scientific) for 1 h at RT. Finally, cells were washed with 1× PBS for 5 min twice and imaged.

### Ellipse fitting algorithm for aspect ratio calculation

To quantify the viscoelastic or viscoplastic behavior of the spheroids, we opted to use aspect ratio of their ejection as a metric. It is obtained by fitting an ellipse to the protruding semi-circular shape coming out of the channel, for four different time points—front when the cluster is 25% exuding out of the channel to when it is almost 75%, on the verge of complete escape (Fig S1).

The ellipse fitting is done using a MATLAB code, to find out the major and minor axes of the hence obtained ellipse. Using those values, we calculated the aspect ratios of the protruding end. (Fig S1). (https://www.mathworks.com/matlabcentral/fileexchange/15125-fitellipse-m). Particle imaging velocimetry was performed using the app PIVLAB on MATLAB (MathWorks, R2023a) (43).

### Modeling framework

CompuCell3D (CC3D) is a problem-solving environment based on the lattice-based GGH (Glazier–Graner–Hogeweg) model or CPM (cellular Potts model) that was designed to model collective behavior of active matter (44). This is done by calculating the Hamiltonian energy function at each simulation step. In the simulation lattice, each cell is represented by rectangular Euclidean lattice sites or pixels that share the same cell ID. The model evolves at each Monte Carlo step (MCS), which consists of index-copy attempts

of each pixel in the cell lattice. Calculation of the Hamiltonian (H) determines the allowed configuration and behavior of cells at each MCS.

$$H = \sum_{i,j\,neighbours} J\big(\tau(\sigma_i), \tau(\sigma_j)\big)\big(1 - \delta(\sigma_i, \sigma_j)\big) + \sum_{\sigma} \big[\lambda_{vol}(v(\sigma_i)$$
$$- V(\sigma_i))^2\big] + \sum_{\sigma} \big[\lambda_{surf}(s(\sigma_i) - S(\sigma_i))^2\big] + H_{EP}.$$

The Hamiltonian used in our model has four main contributors, which are affected by different properties of the cells. The first term in the energy function is the sum over all neighboring pairs of lattice sites $i$ and $j$ with associated contact energies (J) between the pair of cells indexed at those $i$ and $j$. In this term, $i, j$ denotes pixel index, $\sigma$ denotes the cell index or ID, and $\tau$ denotes the cell type. The $\delta$ function ensures that only the $\sigma_i \neq \sigma_j$ terms are calculated ($i, j$ belonging to the same cell will not be considered). Contact energies are symmetric in nature $[J(\tau(\sigma_i),\tau(\sigma_j)) = J(\tau(\sigma_j),\tau(\sigma_i))]$. The contact energy between the two cells is considered inversely proportional to the adhesion between the two cells. The second term in the equation is a function of the volume constraint on the cell. $\lambda_{vol}(\sigma)$ denotes the inverse compressibility of the cell, $v(\sigma)$ is the number of pixels in the cell (*volume*), and $V_t(\sigma)$ is the cell's target volume. The third term in the equation is a function of the surface area constraint on the cell, as the cells have fixed amounts of the cell membrane. For the cell $\sigma$, $\lambda_{surf}(\sigma)$ denotes the inverse membrane compressibility of the cell, $s(\sigma)$ is the surface area of the cell, and $S(\sigma)$ is the cell's target surface area. The fourth term in the equation corresponds to the external potential applied to the center of mass of the cells to cause directional motion. $\Delta H_{EP} = -\overrightarrow{F_{\sigma(i)}}.\overrightarrow{r_{ij}}$ is the Hamiltonian for the external potential for a given MCS. For a cell $\sigma$, during an index-copy attempt from $i$ to $j$, the force vector is $\overrightarrow{F_{\sigma(i)}}$. And the distance between the pixels $i$ and $j$ is $\overrightarrow{r_{ij}}$. The product of these two vectors along the right direction will lead to energy minimization and result in the movement of the cell along that direction.

The acceptance or rejection of the index-copy attempt from pixel $i$ to $j$ depends on the change in the Hamiltonian ($\Delta H$) because of the change in energy after the index-copy attempt. When $\Delta H \leq 0$, the associated index-copy attempt will be successful, and the target pixels will be updated. So, the success probability is $P = 1$. When $\Delta H \geq 0$, the associated index-copy attempt will be successful following the Boltzmann probability function, with a probability of $P = e^{-\left(\frac{\Delta H}{T_m}\right)}$, and it will be unsuccessful with a probability of $P' = 1 - P$. In the Boltzmann probability function, $\Delta H$ represents the calculated change in the overall Hamiltonian of the system between the system configuration at previous MCS and a specific system configuration at the current MCS. $T_m$ relates to the effective membrane fluctuation for the cell and is kept at $T_m = 10$ in all the simulations.

A default dynamical algorithm known as modified Metropolis dynamics with the Boltzmann acceptance function was used at each MCS to move the system toward a low-energy configuration as MCS increases. The term $T_m$ can be considered temperature or magnitude of effective membrane fluctuations. Random movements of the pixels leading to different transition probabilities at each MCS mimic the stochasticity present in biological systems.

## Model components

We used a 450*200*1-pixel square lattice with a non-periodic boundary for all the simulations. Any model element required to participate through MCS pixel-copy attempts must be assigned a cell type. The model consisted of four different cell types: chamber, peripheral cells and core cells, and the medium. All cells were of dimension 5*5*1 in size, except for the core of the moruloids, of size 35*35*1. In an initial configuration, the chamber cells mimic the microfluidic chamber in the experimental setup, with the width-to-length aspect ratio maintained. These cells are "frozen" in the lattice, meaning the motility of the chamber cells is entirely restricted, and the movement of these cells to other lattice positions is not possible. The peripheral cells constituted the outer layer of the spheroids consisting of 32 cells. The core cells constituted the center of the spheroid. In the case of the moruloids, the properties of the core cells were similar to those of the peripheral cells. In the case of blastuloids, the core was made up of a single core cell. The lattices with no assigned cell type or, in other words, the free spaces were assigned as cell type "medium" as a default by the Compu-Cell3D algorithm.

### Contact energies (differential adhesion)

CompuCell3D requires setting interactions between all cell types in the model in terms of contact energies. Contact energy is inversely related to cell–cell adhesion. Higher contact energy between two cell types implies a higher contribution to the effective energy and a lower probability of adhesion of the two cells. Because we have four different cell types, 10 different contact energy values must be assigned. These values were assigned based on previous literature (44, 45). The contact energy values between core–core, peripheral–peripheral, and core–peripheral were considered input variables to observe the effect of morphology on the transition through the chamber (Table 2). In the case of moruloids, core–core, peripheral–peripheral, and core–peripheral values were the same as they all have similar properties.

### External potential

External potential was applied on the core and peripheral cells along the negative x-axis to enable movement of the spheroid through the chamber. After standardizing through parameter sensitization, the value of the force vector was fixed at 6 for all simulations.

### Compressibility

The compressibility of the core cells is different in the moruloids and the blastuloids to model the lumen in blastuloids. In the moruloids, the cells have higher inverse membrane compressibility fixed at 5, similar to the peripheral cells, making the cells stiffer. The core cell in the blastuloids, made to mimic the lumen, has lower inverse membrane compressibility fixed at 0.5 to make them more compressible and fluid-like.

### Modeling deflation and inflation of blastuloids

The blastuloids were modeled to deflate when they met physical resistance while entering the narrow chamber. The velocity of the blastuloids reduced drastically when it met resistance while

**Table 2.** List of different parameter values used in the model.

| Parameter | Value |
|---|---|
| Temperature ($T_m$) | 10 |
| Contact energy parameters | |
| Medium–medium | 10 |
| Medium–core | 10 |
| Medium–peripheral | 10 |
| Medium–chamber | 0 |
| Chamber–chamber | 0 |
| Chamber–core | 50 |
| Chamber–peripheral | 50 |
| Core–core | (0, 5, 10, 15) |
| Core–peripheral | (0, 5, 10, 15) |
| Peripheral–peripheral | (0, 5, 10, 15) |
| Volume and surface area parameters for all cell types (unless specified otherwise) | |
| Target volume | 25 |
| $\lambda_{vol}$ | 5 |
| Target surface area | 20 |
| $\lambda_{surf}$ | 5 |
| Target volume (core of blastuloids) | 400 |
| $\lambda_{vol}$ (core of blastuloids) | 0.5 |
| Target surface area (core of blastuloids) | 80 |
| $\lambda_{surf}$ (core of blastuloids) | 0.5 |

Multiple values assigned for some parameters correspond to the input variables and the values tested.

entering the chamber, and the core of the spheroids was modeled to shrink in size every time the velocity of the spheroid dropped considerably below the baseline velocity. The baseline velocity of these spheroids, averaged over a frequency of 100 MCS, was considered as the velocity at which they moved in the free space because of the constant force acting on them by means of the external potential. When the velocity of the core of the spheroids reduced to less than half its baseline velocity, the target volume and target surface area of the core of the spheroids were reduced, thereby mimicking deflation of the blastuloids. Similarly, on exiting the narrow chamber, the spheroid no longer experiences resistance and inflates. When the velocity of the core of the spheroids increased to twice its baseline velocity, the target volume and target surface area of the core were increased, to model the inflation of the spheroids (https://github.com/MonicaUrd/Spheroid_tunnel_resistance_simulations.git).

## Genetic perturbation

The CDH1 gene shRNA clone was obtained from the MISSION shRNA library (Sigma/Merck). For overexpression, CDH1 cDNA was cloned using PCR from the untransformed MeT-5A line (for primers, see Table 3). The plasmid containing shRNA or scrambled control, or CDH1 cDNA was packaged into lentivirus using packaging vectors

**Table 3. List of PCR forward and reverse primers used in the study.**

| Serial number | Identity | Sequence |
|---|---|---|
| 1 | CDH1 qRT-PCR forward | ACCACCTCCACAGCCACCGT |
| 2 | CDH1 qRT-PCR reverse | GCCCACGCCAAAGTCCTCGG |
| 3 | CDH1 full length forward | ATATCTAGAGCCACCATGGGCCCTTGGAGCCGCAGC |
| 4 | CDH1 full length reverse | ATAGCGGCCGCGTCGTCCTCGCCGCCTCCGTACATG |
| 5 | CDH1 shRNA | CCGGATACCAGAACCTCGAACTATACTCGAGTATAGTTCGAGGTTCTGGTATTTTTTG |
| 6 | 18SrRNA qRT-PCR forward | GTAACCCGTTGAACCCCATT |
| 7 | 18SrRNA qRT-PCR reverse | CCATCCAATCGGTAGTAGCG |

pMD2.G and psPAX2 (packaging vectors were a kind gift from Dr. Deepak K Saini). The plasmids were transfected into 293FT cells (R70007; Thermo Fisher Scientific) using TurboFect (R0533; Thermo Fisher Scientific). Cells were cultured in DMEM supplemented with 10% FBS; conditioned medium containing viral particles was collected at 48 and 72 h. After filtering through a 0.45-$\mu$m filter, viral particles were concentrated using the Lenti-X concentrator as mentioned in the manufacturer's protocol (631232; TaKaRa). The concentrated virus was aliquoted and stored at –80°C until use. Cells were seeded in a 24-well plate at 50–60% confluence and transduced with viral particles containing shRNA or scrambled control along with polybrene (4 $\mu$g/ml) for 24 h. After 72 h, transduced cells were selected using 5 $\mu$g/ml puromycin (CMS8861; HiMedia). The knockdown of the gene was assayed using real-time PCR. Selection for the perturbation was performed by exposure for up to 3 $\mu$g/ml puromycin over 3–4 passages in order to ensure homogeneity in overexpression or knockdown.

### Microfluidic chip fabrication

A Piranha-cleaned 4″ silicon wafer was spin-coated with SU-2035 (Microchem) negative photoresist at 511$g$ for 35 s to obtain a thickness of 75 $\mu$m. After a soft bake for 3 min at 65°C, it was UV-exposed using an MJB-4 optical lithography tool through a 5″ photomask containing the micropatterns. The wafer was then developed using SU-8 developer for 15 min, followed by an IPA wash. It was coated with Teflon followed by a hard bake for 45 min at 150°C. The pattern was transferred to a glass slide using soft lithography. PDMS was mixed with a curing agent in a 10:1 ratio and poured onto the wafer with a mold. It was desiccated in vacuum for around 5 min to remove any bubbles. After that, it was heated for 45 min on 110°C, and the hardened mold was peeled off once it was cooled. Holes were punched for inlets and outlets using a biopsy punch, after which the PDMS block was plasma-bonded to an IPA cleaned glass slide after 1 min of treatment. The device was heated for 15 min at 110°C to create a permanent bond.

### Experimental setup

The sample was loaded into the device using a BD 1-ml syringe and microtubing. Using Chemyx Fusion high-pressure pump, it was withdrawn (to avoid settling of clusters), at a rate of 1,000 $\mu$l/hr. The flow was visualized and recorded using a high-speed camera at 1,000 fps, with 10x (for the single channel) and 4X (for the 2-channel design) magnification lenses. Photron FASTCAM Viewer (PFV-4) was used to view and analyze recorded videos. In the videos, 1 $\mu$m corresponds to 1 pixel.

## Availability of Data and Material

The raw data for all experiments and simulations will be made available upon reasonable request.

## Supplementary Information

## Acknowledgements

We would like to acknowledge Shahid Hussain and Anchita Gopikrishnan for help with spheroid preparation. This work was supported by the Wellcome Trust/DBT India Alliance Fellowship grant (IA/I/17/2/503312) awarded to R Bhat. It was also supported by the John Templeton Foundation (#62220), by the Department of Biotechnology, India (BT/909 PR26526/GET/119/92/2017 and BT/PR21962/NNT/28/1233/2017), and by the Indo-French Centre for the Promotion of Advanced Research (69T08-2) to R Bhat. P Sen acknowledges the Department of Biotechnology (DBT) and the Ministry of Electronics and Information Technology (MeitY). The opinions expressed in this study are those of the authors and not those of the John Templeton Foundation. J Langthasa and T Dutt acknowledge the Indian Institute of Science (IISc) for fellowship. M Umesh and S Bothra acknowledge KVPY for the student scholarship (X08010248, X113030157).

### Author Contributions

T Dutt: formal analysis, investigation, and methodology.
J Langthasa: formal analysis, investigation, and methodology.
M Umesh: software.
S Mishra: formal analysis and investigation.
S Bothra: formal analysis and investigation.
K Vidhipriya: investigation, methodology, and writing—review and editing.
A Vadaparty: resources and data curation.

P Sen: resources, data curation, formal analysis, investigation, and methodology.

R Bhat: conceptualization, resources, data curation, formal analysis, validation, investigation, visualization, and methodology.

## Conflict of Interest Statement

The authors declare that they have no conflict of interest.

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
