## [Reviewer comments · Life Science Alliance]

Life Science Alliance

Rheological transition driven by matrix makes cancer spheroids resilient under confinement

Tavishi Dutt, Jimpi Langthasa, Monica Umesh, Satyarthi Mishra, Siddharth Bothra, Kotpalli Vidhipriya, Annapurna Vadaparty, Prosenjit Sen, and Ramray Bhat

DOI: <https://doi.org/10.26508/lsa.202402601>

Corresponding author(s): Ramray Bhat, Indian Institute of Science Bangalore and Prosenjit Sen, Indian Institute of Science

Review Timeline:	Submission Date:	2024-01-18
	Editorial Decision:	2024-03-28
	Revision Received:	2025-01-31
	Editorial Decision:	2025-02-18
	Revision Received:	2025-03-02
	Accepted:	2025-03-03

Transaction Report:

March 28, 2024

Re: Life Science Alliance manuscript #LSA-2024-02601-T

Prof. Ramray Bhat
Indian Institute of Science Bangalore
Department of Molecular Reproduction, Development and Genetics
GA07 New Biological Sciences Building
Bangalore, Karnataka 560012
India

Dear Dr. Bhat,

Thank you for submitting your manuscript entitled "Rheological transition driven by matrix makes cancer spheroids resilient under confinement". The manuscript has been evaluated by expert reviewers, whose reports are appended below. Unfortunately, after an assessment of the reviewer feedback, our editorial decision is against publication in Life Science Alliance.

Although your manuscript is intriguing, I feel that the points raised by the reviewers are more substantial than can be addressed in a typical revision period. If you wish to expedite publication of the current data, it may be best to pursue publication at another journal.

Given the interest in the topic, I would be open to re-submission to Life Science Alliance of a significantly revised and extended manuscript that fully addresses the reviewers' concerns and is subject to further peer review. If you would like to resubmit this work to Life Science Alliance, you may submit an appeal directly through our manuscript submission system. Please note that priority and novelty would be reassessed at re-submission.

Regardless of how you choose to proceed, we hope that the comments below will prove constructive as your work progresses.

Thank you for thinking of Life Science Alliance as an appropriate place to publish your work.

Sincerely,

Reviewer #1 (Comments to the Authors (Required)):

Summary of the paper:

Metastatic cancer cells migrate as clusters, referred here as spheroids, and have to face various physical constraints imposed by their environment. They obviously overcome these constraints as demonstrated by the lethal outcome of cancer metastasis. How do they succeed? The authors address this issue from the original point of view of analysing the mechanical properties of human ovarian cancer cell spheroids. In this in vitro model, the cancer cells first form moruloid-like spheres that later change their morphology into blastuloid-like spheres. Blastuloids are characterized by the presence of a lumen and an ECM coat, both absent in moruloids. Using a microfluidic platform to spatially constrain the spheroids, the authors analyse the mechanical properties of the two types of spheroids and show that they display very different behaviours. In short, blastuloids are more resistant to physical constraints than moruloids and this resilience, as called by the authors, relies on the presence of a lumen and of an external basal membrane, also referred to as ECM by the authors. The authors also conclude that their data indicate that the transition of moruloid to blastuloid occurs concomitantly to a mesenchymal to epithelial transition, based on the high expression of E-Cadherin in blastuloids and on their computational modelling. With this study, the authors showcase (I cite) "how a limited set of proteins (those of the ECM associated to the blastuloids) shift multicellular states both in terms of their polarity and motility."

In summary this is a very interesting study for a broad range of scientists which however would gain in robustness if some experimental aspects would be addressed more carefully.

Data:

Data provided in the manuscript convincingly support the following points:

- blastuloids have a higher elastic modulus than moruloids, travel faster through the microfluidic channels, undergo less disintegration and recover faster their shape after exiting the channels.
- the level of expression of E-Cadherin does not impact on the mechanical behaviour of the spheroids
- on the opposite, the presence of a protein coat on the spheroid is crucial for its resilience and dictates its mechanical behaviour.

Less convincing are the following sets of data:

- 1-At the end of the section describing the results presented in Figure 4, the authors state that "Upon ECM degradation, cavitation is lost". As stated below (Additional issues, point 4), I cannot see a loss in cavity since I cannot see the cavity first. Please provide qualitative (better images) data and quantitative data: how reproducible is this loss of cavity? Is it observed for each blastuloid treated, for a few percent ?
- 2-Data provided in the manuscript on which the authors rely to state that the spheroids undergo a mesenchymal to epithelial transition when they transit from moruloids to blastuloids. If I understand correctly, the authors base this statement on the higher expression of E-Cadherin by the blastuloids and on the observation (not shown and not referenced in the paper) of fibrinogen expression by moruloids. A high level of E-Cadherin expression is certainly typical for epithelial cells but one marker does not make the identity of a cell. Same holds true for mesenchymal cells. In view of the complexity of the EMT and MET events, I would advise the authors to either smoothen or discuss this statement more thoroughly since it is an interesting issue, or to perform further experiments to consolidate their statement. I would recommend the reading of the following review on that topic: Yang, J., Antin, P., Bex, G. et al. Guidelines and definitions for research on epithelial-mesenchymal transition. *Nat Rev Mol Cell Biol* 21, 341-352 (2020). <https://doi.org/10.1038/s41580-020-0237-9>. The authors could check for the expression of more proteins specific of both states (mesenchymal and epithelial) in blastuloids and moruloids, either at the gene level (RT-qPCR) or protein level (western blot if they pool spheroids to get enough material or IHC). I do not know how long it takes to generate both types of spheroids (this information is missing in the manuscript), but I believe that, if the authors decide to perform further experiments, they could manage a resubmission within 6 months.

Additional issues:

1-Abstract:

Taking into account my remarks under Data 1 and 2, the two last sentences of the Abstract should be carefully rephrased. Currently, the authors state: "Although, E-cadherin overexpression in moruloids did not affect their resilience, blastuloid ECM-debridement decreased E-cadherin membrane localization, obliterated the lumen, and reversed the rheological properties of blastuloids to those typifying moruloids. The ECM-induced lumen therefore drives spheroidal transition from a labile viscoplastic to a resilient elastic state allowing them to survive spatially-constrained peritoneal flows." The authors need to provide more convincing data for the "obliteration of the lumen by ECM degradation" and I could not find the data supporting the "induction of the lumen by ECM".

2-Material and Methods:

Please provide information on the generation of the moruloids and blastuloids with the human ovarian cell lines (number of cells, time for generation for each type of spheroids?).

The transfection of the cells is not described: is the transfection done at the level of the cells, or on the spheroids? What is the efficiency of transfection (percent of cells expressing more E-Cad ?) Does it affect the generation of the spheroids ? Please provide this information.

The labelling with F-phalloidin is not described, please add this description.

How many independent experiments were conducted for each type of analysis?

3-Comments related to Figure 3 and its text:

Cell detachment, spheroidal damage and disintegration: the authors clearly show an example of cell detachment in Figure 3C, middle image. I would advise them to give a clear example of a damaged spheroid and of disintegration, so that the reader understands what the authors mean with spheroidal damage and spheroidal disintegration.

Legend to Figure 3: scale bars are mentioned but I cannot see scale bars in the Figure 3. Please correct.

4-Comments related to Figure 4 and its text:

Legend to Figure 4:

There are two (F) and the second one should be for (G). Please correct.

The last sentence reads: "Scale bars = 50 um and 20 um respectively". Respective of what? The scales should rather be referred to the panel they belong to:(H) and (I). Please correct.

Figure 4I: where is the cavity in the control blastuloid? It is clear that collagenase-treated spheroids present a morphology that exclude a cavity, but I could say the same for the non-treated spheroids. The authors should indicate where it is or show it on another -and convincing- example of spheroids with cavity.

Text:

The authors should explain at this step of their report what is the purpose of using collagenase.

5-Supplementary pdf files for Figures S5 and S6: the files seem to be incomplete and they are of bad quality. Please check, correct and put Figure 5 before Figure 6 in the corrected version.

6-Discussion:

The authors state: "Taken together, the blastuloid-moruloid shift corresponds to a mesenchymal-to-epithelial transition (MET)." The way the sentence is currently written implies that blastuloids are of mesenchymal nature and moruloids of epithelial nature. If I correctly understood the paper, the authors want to say that the moruloid to blastuloid transition corresponds to a mesenchymal to epithelial transition. The authors should thus rephrase this sentence.

Reviewer #2 (Comments to the Authors (Required)):

This manuscript investigates the migratory behavior of two types of morphologically distinct ovarian cancer spheroids under confining conditions. This is an interesting and relevant topic since understanding the behavior of such spheroids may eventually be an avenue towards specific therapeutic approaches. The design of the flowcell experiment is an interesting and tractable approach to study some of those aspects. Unfortunately several of the experiments are quite poorly controlled and statements on the involved mechanisms of spheroid migration and specific survival are not supported by the results as presented in this manuscript, and this reduces my enthusiasm to a great extent. As this reviewer does not have sufficient background in mathematical modelling, no specific comments are given on the execution of those parts of the manuscript.

Main points:

The analysis of the intercellular rearrangements as shown in Figure 3 A and B, and the conclusion that moruloids undergo more cellular rearrangements and jamming-unjamming transitions are confusing and appear to rely on an improper comparison. In A1 the areas analysed in moruloids are depicted by A1 and A2. What areas are analysed in the blastuloids? Since the lumen is empty in those structures, there are no cells to analyze and thus it seems no surprise that the vector maps do not change much. Since a large part of the manuscript builds on the conclusion that moruloids undergo more unjamming, the cellular rearrangement analysis should be expanded and at least comprise comparable cell-containing areas of the cellular assemblies, and thus specifically the periphery of the structures.

In figure 4 and 5, experiments were aimed at understanding the role of E-cadherin-based cell-cell adhesion and the presence of ECM in the blastuloid and moruloid behavior. The rationale of some of these (wet-lab) experiments is not always clear, and the experiments are not well controlled, leading to hard to interpret results.

There are several issues:

- First the authors introduce this topic by stating that blastuloids have higher basement membrane expression based on earlier work that is not cited, and is therefore difficult to place in context considering that the basement membrane contains many different proteins. Next, they claim, again without giving references, that laminin111 regulates E-cadherin expression, and use this information to introduce why they will investigate E-cadherin expression. There are many good arguments to hypothesize higher E-cadherin expression in blastuloids, not in the least from embryogenesis, where it is well known that E-cadherin null embryos can form morulas but will die in the blastula stage and thus the arguments above are quite weak.
- Second, the images of E-cadherin staining in moruloids and blastuloids in Figures 4I and 4H cannot be compared since they are of different magnification and different colors are being used suggesting that these images do not come from a direct controlled comparison.
- Third, the role of E-cadherin seems to be analyzed by degrading the ECM in blastuloids. Indeed, there is a decrease in E-cadherin staining, but this is a very indirect effect that cannot be directly attributed to a loss of function of E-cadherin, but rather a loss-of-function of ECM. Furthermore, the experimental conditions by which the ECM is degraded is quite extreme, both in terms of time and amount of collagenase used. Since most collagenases are isolated from bacteria, they tend to be quite crude and contain many different proteases beyond collagenase that can degrade many membrane proteins, and thus it is not a surprise that the blastuloids lose their integrity. If the authors want to study a loss-of-function of E-cadherin, then some knockout or knockdown studies should be performed.
- Finally, despite that introduction of Figure 5, that starts with the sentence: "To verify if an increased intercellular adhesion driven by higher E-cadherin localization contributed to the mechanical behavior of blastuloid spheroids.....", the authors in fact show that E-cadherin is not involved in the different migration kinetics, but that ECM expression is the determining factor. However, apart from the collagenase experiments, which have the above-mentioned issues, no experiments (either wet-lab or in silico) are specifically devoted to this issue.

In all, while understanding the different behaviors of the two tumor cell assemblies is relevant, the issues above makes this a quite incomplete study in which the conclusions are not sufficiently supported by the presented results.

Reviewer #1 (Comments to the Authors (Required)):

Summary of the paper:

Metastatic cancer cells migrate as clusters, referred here as spheroids, and have to face various physical constraints imposed by their environment. They obviously overcome these constraints as demonstrated by the lethal outcome of cancer metastasis. How do they succeed? The authors address this issue from the original point of view of analysing the mechanical properties of human ovarian cancer cell spheroids. In this in vitro model, the cancer cells first form moruloid-like spheres that later change their morphology into blastuloid-like spheres. Blastuloids are characterized by the presence of a lumen and an ECM coat, both absent in moruloids. Using a microfluidic platform to spatially constrain the spheroids, the authors analyse the mechanical properties of the two types of spheroids. and show that they display very different behaviours. In short, blastuloids are more resistant to physical constraints than moruloids and this resilience, as called by the authors, relies on the presence of a lumen and of an external basal membrane, also referred to as ECM by the authors. The authors also conclude that their data indicate that the transition of moruloid to blastuloid occurs concomitantly to a mesenchymal to epithelial transition, based on the high expression of E-Cadherin in blastuloids and on their computational modelling. With this study, the authors showcase (I cite) "how a limited set of proteins (those of the ECM associated to the blastuloids) shift multicellular states both in terms of their polarity and motility."

In summary this is a very interesting study for a broad range of scientists which however would gain in robustness if some experimental aspects would be addressed more carefully.

Data:

Data provided in the manuscript convincingly support the following points:

- blastuloids have a higher elastic modulus than moruloids, travel faster through the microfluidic channels, undergo less disintegration and recover faster their shape after exiting the channels.
- the level of expression of E-Cadherin does not impact on the mechanical behaviour of the spheroids
- on the opposite, the presence of a protein coat on the spheroid is crucial for its resilience and dictates its mechanical behaviour.

Less convincing are the following sets of data:

1.1 At the end of the section describing the results presented in Figure 4, the authors state that "Upon ECM degradation, cavitation is lost". As stated below (Additional issues, point 4), I cannot see a loss in cavity since I cannot see the cavity first. Please provide qualitative (better images) data and quantitative data: how reproducible is this loss of cavity? Is it observed for each blastuloid treated, for a few percent?

Explanation:

We thank the reviewer for raising this concern. The formation of lumen was the key finding of a paper that preceded this manuscript and was incidentally also published in Life Science Alliance (Langthasa et al, Lif Sci All, 2021). In this paper, we showed how the debridement of the ECM by collagenase resulted in the loss of lumen in blastuloids; in fact, using time lapse videomicroscopy, we established temporal kinetics not just for lumen loss but also recovery of lumen upon washing away the collagenase (see below Figure from paper).

(G) Bright-field photomicrographs taken at 0, 16, and 48 h from time-lapse videography of blastuloid OVCAR3 spheroids initiated after addition of collagenase IV (see Video S6). (G) White dotted lines in the black background highlight the changes in the contour of lumen in (G). (three independent repeats with multiple spheroids analyzed for each repeat) Graph on the right shows change in lumen size calculated using paired t test. Bars represent mean \pm SD from a representative experiment. (H) Bright-field photomicrographs taken at 0, 6, and 18 h from time-lapse videography of OVCAR3 spheroids pretreated with Collagenase IV with videography initiated after the removal of Collagenase IV (see Video S7). (H) White dotted lines in the black background highlight the changes in the contour of lumen in (H). (n = 3 independent repeats with multiple spheroids analyzed for each repeat) Graph on the right shows change in lumen size calculated using paired t test. Bars represent mean \pm SD from a representative experiment.

Most of the moruloids (75-90%) cultured over time form blastuloids and when collagenase treatment is given, most (75-90%) of those that formed blastuloids, lose their lumen. The loss in lumen is spectacularly evidenced in brightfield as the spheroids do not have to be subjected to fixation and staining with multiple rounds of washes, which could damage or distort the cavitation architecture of the spheroids.

Revision in the manuscript:

Keeping the reviewer's point in mind and bridging the relevant findings from our previous work with the current paper, we are providing supplementary images showing the clear loss of lumen in spheroids

upon Collagenase treatment (see image right compared with control (left)). This will help the current readers make the connection better without having to go through past papers. We

also provide the above statistical detail of conversions for better understanding of the readers.

1.2-Data provided in the manuscript on which the authors rely to state that the spheroids undergo a mesenchymal to epithelial transition when they transit from moruloids to blastuloids. If I understand correctly, the authors base this statement on the higher expression of E-Cadherin by the blastuloids and on the observation (not shown and not referenced in the paper) of fibrinogen expression by moruloids. A high level of E-Cadherin expression is certainly typical for epithelial cells but one marker does not make the identity of a cell. Same holds true for mesenchymal cells. In view of the complexity of the EMT and MET events, I would advise the authors to either smoothen or discuss this statement more thoroughly since it is an interesting issue, or to perform further experiments to consolidate their statement. I would recommend the reading of the following review on that topic: Yang, J., Antin, P., Berx, G. et al. Guidelines and definitions for research on epithelial-mesenchymal transition. *Nat Rev Mol Cell Biol* 21, 341-352 (2020). <https://doi.org/10.1038/s41580-020-0237-9>. The authors could check for the expression of more proteins specific of both states (mesenchymal and epithelial) in blastuloids and moruloids, either at the gene level (RT-qPCR) or protein level (western blot if they pool spheroids to get enough material or IHC). I do not know how long it takes to generate both types of spheroids (this information is missing in the manuscript), but I believe that, if the authors decide to perform further experiments, they could manage a resubmission within 6 months.

Explanation:

The reviewer raises a very valid point. We have incorporated a strongly histological understanding of epithelialness or mesenchymality. By this, we mean epithelialness to be the ability to form

polygonal jammed states of cells localized on an ECM sheet and being sessile with strong cell-cell adhesion through junctional proteins as is evident from cells within blastuloids. By mesenchymality, we mean the ability of cells to deform their shape temporally, lose cell-cell

adhesion, secrete fibronectin and migrate with respect to each other, as seen in moruloids. In our earlier publication, we had rigorously shown several properties of mesenchymality and epithelialness in moruloids and blastuloids, such as the ability of cells within moruloids to move past each other within spheroids whereas in blastuloids, the inter-cell relationships are fixed (see above). In addition, we also established the presence of tight junctions within blastuloids through the correct localization of Zonula occludens-1 (ZO-1) and occludin and the cortical localization of Ezrin (see above). These indicate an epithelioid morphological phenotype that can be ascribed to blastuloids.

Revision in the manuscript: We describe all our published data explained above from our previous paper with proper referencing to indicate why we refer to a mesenchymal to epithelioid transition (please note the change from epithelial to epithelioid given that these cancerous cells can never be truly epithelial which has constraints of classic histological definitions). We also include the references provided by the reviewer in our reframing.

Additional issues:

1.3-Abstract:

Taking into account my remarks under Data 1 and 2, the two last sentences of the Abstract should be carefully rephrased. Currently, the authors state: "Although, E-cadherin overexpression in moruloids did not affect their resilience, blastuloid ECM-debridement decreased E-cadherin membrane localization, obliterated the lumen, and reversed the rheological properties of blastuloids to those typifying moruloids. The ECM-induced lumen therefore drives spheroidal transition from a labile viscoplastic to a resilient elastic state allowing them to survive spatially-constrained peritoneal flows."

The authors need to provide more convincing data for the "obliteration of the lumen by ECM degradation" and I could not find the data supporting the "induction of the lumen by ECM".

Revision in the manuscript

We now include more convincing data for the first point (see above) in the revised manuscript. For the second, removal of the basement membrane (BM) using collagenase

results in loss of lumen, whereas removal of the collagenase restored the lumen (see above). We now specifically refer to these findings in the revised manuscript. In addition, we have

in response to the reviewer's point performed the experiment in which we cultivate spheroids in serum-free conditions with and without supplementation with laminin-rich ECM (4%). We observe that spheroids that been formed in the presence of basement membrane ECM form, lumen-like compacted structures to a greater extent within a period of time in which non BM supplemented clusters are uncompacted with no lumen, validating our proposition.

1.4-Material and Methods:

Please provide information on the generation of the moruloids and blastuloids with the human ovarian cell lines (number of cells, time for generation for each type of spheroids?).

Revision in the manuscript:

The revised manuscript now has a detailed section on the formation of moruloids and blastuloids with all the information the reviewer has requested. We also add references to specific sections from our previous paper (Langthasa et al, 2021) which has all these details as well.

1.5 The transfection of the cells is not described: is the transfection done at the level of the cells, or on the spheroids? What is the efficiency of transfection (percent of cells expressing more E-Cad ?) Does it affect the generation of the spheroids ? Please provide this information.

Explanation: We stably transduced E-Cadherin (and now, E-cadherin shRNA) in OVCAR-3 cells using lentiviral cDNA or shRNA delivery in order to achieve efficient overexpression (and now knockdown). Selection for the perturbation was performed by exposure for upto 3 µg/mL puromycin over 3-4 passages in order to ensure homogeneity in overexpression or knockdown. In addition, we have in an experiment not relating to the manuscript, overexpressed a E-cadherin YFP construct using the same transduction protocol where we see the fluorescence in all the cells.

Revision in the manuscript:

The methods section of the revised manuscript incorporates a detailed explanation of the mechanism of transduced overexpression and knockdown. It also incorporates the data showing the extent of overexpression and knockdown and data on the efficiency of spheroidogenesis which is linked to its mechanical properties.

1.6 The labelling with F-phalloidin is not described, please add this description.

This has now been incorporated into the revised manuscript.

1.7 How many independent experiments were conducted for each type of analysis?

The figure legends mention the number of replicates for each experiment. In addition, this information is now also provided in the methods section on statistics.

1.8 3-Comments related to Figure 3 and its text:

Cell detachment, spheroidal damage and disintegration: the authors clearly show an example of cell detachment in Figure 3C, middle image. I would advise them to give a clear example of a damaged spheroid and of disintegration, so that the reader understands what the authors mean with spheroidal damage and spheroidal disintegration.

Revision in the manuscript:

Thank you. We now have in our revised manuscript version a fast speed imaging time panel of clear evidence of what we mean by disintegration, where cells (shown by yellow arrowheads) can be seen getting detached from moruloids as they enter and exit through the confinement channel.

1.9 Legend to Figure 3: scale bars are mentioned but I cannot see scale bars in the Figure 3. Please correct.

In the revised manuscript version, this has been corrected.

1.10 4-Comments related to Figure 4 and its text:

Legend to Figure 4:

There are two (F) and the second one should be for (G). Please correct.

The last sentence reads: "Scale bars = 50 μ m and 20 μ m respectively". Respective of what? The scales should rather be referred to the panel they belong to:(H) and (I). Please correct.

These are corrected, thank you.

1.11 Figure 4I: where is the cavity in the control blastuloid? It is clear that collagenase-treated spheroids present a morphology that exclude a cavity, but I could say the same for the non-treated spheroids. The authors should indicate where it is or show it on another -and convincing-example of spheroids with cavity.

The revised manuscript has a replaced figure with a lumenized blastuloid, thank you

1.12 Text:

The authors should explain at this step of their report what is the purpose of using collagenase.

The revised manuscript provides a detailed description of why the collagenase treatment is key.

1.13 5-Supplementary pdf files for Figures S5 and S6: the files seem to be incomplete and they are of bad quality. Please check, correct and put Figure 5 before Figure 6 in the corrected version.

The revised version correctly presents this data

1.14 6-Discussion:

The authors state: "Taken together, the blastuloid-moruloid shift corresponds to a mesenchymal-to-epithelial transition (MET)." The way the sentence is currently written implies that blastuloid are of mesenchymal nature and moruloids of epithelial nature. If I correctly understood the paper,

the authors want to say that the moruloid to blastuloid transition corresponds to a mesenchymal to epithelial transition. The authors should thus rephrase this sentence.

The current revised manuscript has corrected the sentence, the reviewer interpreted it correctly. However, we exercise due caution even though we have more markers shown in the rebuttal and the revised manuscript. Please see our specific response to point 1. 2

Reviewer #2 (Comments to the Authors (Required)):

This manuscript investigates the migratory behavior of two types of morphologically distinct ovarian cancer spheroids under confining conditions. This is an interesting and relevant topic since understanding the behavior of such spheroids may eventually be an avenue towards specific therapeutic approaches. The design of the flowcell experiment is an interesting and tractable approach to study some of those aspects. Unfortunately several of the experiments are quite poorly controlled and statements on the involved mechanisms of spheroid migration and specific survival are not supported by the results as presented in this manuscript, and this reduces my enthusiasm to a great extent. As this reviewer does not have sufficient background in mathematical modelling, no specific comments are given on the execution of those parts of the manuscript.

Main points:

2.1 The analysis of the intercellular rearrangements as shown in Figure 3 A and B, and the conclusion that moruloids undergo more cellular rearrangements and jamming-unjamming transitions are confusing and appear to rely on an improper comparison. In Ai the areas analysed in moruloids are depicted by A1 and A2. What areas are analysed in the blastuloids? Since the lumen is empty in those structures, there are no cells to analyze and thus it seems no surprise that the vector maps do not change much. Since a large part of the manuscript builds on the conclusion that moruloids undergo more unjamming, the cellular rearrangement analysis should be expanded and at least comprise comparable cell-containing areas of the cellular assemblies, and thus specifically the periphery of the structures.

Explanation:

We thank the reviewer for pointing out. It took us a lot of effort to resolve this issue, but we have we believe clearly established an analytical pipeline for our PIV observations that completely differs from what we had in the previous version.

Revision in the manuscript:

In the revised manuscript, we use specific ROIs that are only of the cellular portion of the blastuloids and measure the mean flow angle of this ROI (green dotted rings). The ROI size is kept similar and 4 ROIs taken for each spheroid (at the back, front, top and bottom). We then compare the mean flow vector angle (relative to the direction of motion) across 5 spheroids. Our results show that this metric shows a greater mean and variation in the moruloids, suggesting different cell clusters are moving with greater variation with respect to each other in moruloids relative to blastuloids. The metric has also been applied to show how loss in BM ECM due to debridement as well as E-cadherin KD leads to a similar attenuation in morphogenetic resilience in clusters

compared to blastuloids.

In figure 4 and 5, experiments were aimed at understanding the role of E-cadherin-based cell-cell adhesion and the presence of ECM in the blastuloid and moruloid behavior. The rationale of some of these (wet-lab) experiments is not always clear, and the experiments are not well controlled, leading to hard to interpret results.

There are several issues:

2.3 - First the authors introduce this topic by stating that blastuloids have higher basement membrane expression based on earlier work that is not cited and is therefore difficult to place in context considering that the basement membrane contains many different proteins.

In the revised version of the manuscript, we now have properly cited the reference to the precursor published paper from our group which establishes the expression and function of basement membrane (laminin, Collagen IV as well as fibulin) in the context of blastuloids. In short, the basement membrane gives the spheroids a morphological integrity by preventing cells from further adding on to an already formed spheroid as well as preventing spheroidal coalescence. In addition, it decreases the ability of spheroids to attach on to peritoneal surfaces. What we had not established in the previous paper was whether the morphological stability of blastuloid was also correlated with mechanical stability. This paper establishes that. We note this point in the discussion in a clearer fashion.

Next, they claim, again without giving references, that laminin111 regulates E-cadherin expression, and use this information to introduce why they will investigate E-cadherin expression. There are many good arguments to hypothesize higher E-cadherin expression in blastuloids, not in the least from embryogenesis, where it is well known that E-cadherin null embryos can form morulas but will die in the blastula stage and thus the arguments above are quite weak.

We apologize for this oversight. Please see now the paragraph, inserted in the revised manuscript, which provides copious references for the setting up of the hypothesis on how basement membrane is responsible for E-cadherin expression.

“Interepithelial adhesion is principally mediated through basolateral adherens junctions, established using homodimeric interactions between transmembrane cell adhesion molecules such as E-cadherin. The expression of E-cadherin has been

shown to be under the regulation of Laminin 111, a principal constituent of epithelial basement membrane (BM) matrix: the latter decreases DNMT1 in breast cells resulting in a decrease in promoter methylation of CDH1 gene that encodes for E-cadherin (Benton et al, FASEB J, 2009). In fact, work across several systems have established an instructive role of basement membrane matrix in establishing the apicobasal polarity as is evidenced in blastuloids (e.g., in MDCK cysts, kidney and lung organotypic cultures and murine embryoid bodies)(O'Brien et al, Nat Cell Bio, 2001; Klein et al, Cell, 1988; Schuger et al, Int J Dev Biol, 1998; Murray et al, J Cell Biol, 2000). We have demonstrated in an earlier paper that an increased expression and relocalization of BM-typical proteins such as fibulin, Collagen IV and laminins are associated with blastuloid formation.”

2.4 - Second, the images of E-cadherin staining in moruloids and blastuloids in Figures 4I and 4H cannot be compared since they are of different magnification and different colors are being used suggesting that these images do not come from a direct controlled comparison.

Explanation:

We respectfully submit to the reviewer that these two are separate experiments and they are not meant to be compared. The first experiment (4H) shows how E-Cadherin levels are higher in blastuloids (right) compared with moruloids (left). The imaging is done using maximum intensity projection in order to infer the level of expression across the spheroid.

The second experiment shows blastuloid spheroids wherein basement membrane has been debrided using Type IV collagenase (right) wherein E-cadherin is delocalized and depleted compared with control (right), with the imaging done for an intermediate Z section.

Revision in the manuscript:

We have now reversed the colors for the images. This will create considerably less confusion.

2.5 - Third, the role of E-cadherin seems to be analyzed by degrading the ECM in blastuloids. Indeed, there is a decrease in E-cadherin staining, but this is a very indirect effect that cannot be directly attributed to a loss of function of E-cadherin, but rather a loss-of-function of ECM. Furthermore, the experimental conditions by which the ECM is degraded is quite extreme, both in terms of time and amount of collagenase used. Since most collagenases are isolated from bacteria, they tend to be quite crude and contain many different proteases beyond collagenase that can degrade many membrane proteins, and thus is it not a surprise that the blastuloids loose their integrity. If the authors want to study a loss-of-function of E-cadherin, then some knockout or knockdown studies should be performed.

Explanation:

The reviewer raises several pertinent points which admittedly deserve attention. In relation to the collagenase, we have conducted the experiments using Type IV collagenase since we share the same apprehensions as the reviewer: Type IV collagenase is far purer than other (Type I-III) collagenases which have non specificity and the limitations highlighted by the reviewer. Type IV

collagenase is known to have the lowest non-collagenase proteolytic activity leaving the membrane proteins intact. In fact, we performed appropriate experiments in our previous manuscript, wherein the dose and the timing was optimized to give results that are specific to spheroidogenesis without impairing cell viability: this to the extent that when washed away, the spheroids regain function in terms of lumen formation. This description is now part of the appropriate methods section in the revised manuscript.

Revision in the manuscript:

We agree with the reviewer that knockdown of E-cadherin would be a useful experiment to perform and validate our phenotype. We therefore performed the same. In the revised manuscript, we show that knocking down E-cadherin decelerated lumen formation with respect to controls. In addition, entry time was now better correlated with size and they showed a greater degree of cellular detachment and damage relative to blastuloid controls. Therefore E-cadherin knockdown phenocopies to an extent, the effect of basement membrane removal on spheroidal morphogenesis. Please see also our explanation to the last point of the reviewer.

2.6 - Finally, despite that introduction of Figure 5, that starts with the sentence: "To verify if an increased intercellular adhesion driven by higher E-cadherin localization contributed to the mechanical behavior of blastuloid spheroids.....", the authors in fact show that E-cadherin is not involved in the different migration kinetics, but that ECM expression is the determining factor. However, apart from the collagenase experiments, which have the above-mentioned issues, no experiments (either wet-lab or in silico) are specifically devoted to this issue.

In response to the formidable set of suggestions provided by the reviewers, especially the E-cadherin knockdown experiments, which we have now done, our story is now modified: we posit that E-cadherin whose localization is driven through the basement membrane matrix is involved in spheroidal integrity although it does not affect the temporal dynamics of the transit. These results came from our analysis of the E-cadherin-depleted spheroids. However, E-cadherin KD results in greater spheroidal disintegration and an increased asymmetry in intraspheroidal cell flows and abrogation of spheroidal lumen. These results are now part of our manuscript (Figure 6). In short, both basement membrane-debrided and E-cadherin-depleted blastuloid spheroids phenocopy moruloid mechanical behavior. The debridement experiments produces a more drastic phenotype in terms of effect on kinetics because its effects are manifest on spheroids that were formed normally till

the blastuloid state and subsequently debrided, as opposed to E-cadherin knockdown which was operative in cells even before they were suspended for spheroid formation. The debridement-analogous experiment would be an induced knockdown of E-cadherin post moruloid formation, which is challenging given the time window of moruloid-to blastuloid transition. We discuss this limitation in the discussion section of our revised manuscript.

In all, while understanding the different behaviors of the two two tumor cell assemblies is relevant, the issues above makes this a quite incomplete study in which the conclusions are not sufficiently supported by the presented results.

February 18, 2025

RE: Life Science Alliance Manuscript #LSA-2024-02601-TR-A

Prof. Ramray Bhat
Indian Institute of Science Bangalore
Department of Developmental Biology and Genetics
GA07 New Biological Sciences Building
Bangalore, Karnataka 560012
India

Dear Dr. Bhat,

Thank you for submitting your revised manuscript entitled "Rheological transition driven by matrix makes cancer spheroids resilient under confinement". We would be happy to publish your paper in Life Science Alliance pending final revisions necessary to meet our formatting guidelines.

- please be sure that the authorship listing and order is correct
- please upload all figure files as individual ones, including the supplementary figure files; all figure legends should only appear in the main manuscript file
- please add ORCID ID for the secondary corresponding author -- they should have received instructions on how to do so
- please add the Twitter/X and Bluesky handles of your host institute/organization as well as your own or/and one of the authors in our system
- please upload your Tables in editable .doc or excel format
- please add your main, supplementary figure, and table legends to the main manuscript text after the references section
- there is a call-out for Figure S9 and this figure has not been provided
- please add callouts for Figures 4B, C, and S8 to your main manuscript text

LSA now encourages authors to provide a 30-60 second video where the study is briefly explained. We will use these videos on social media to promote the published paper and the presenting author (for examples, see <https://docs.google.com/document/d/1-UWCfbE4pGcDdcgzcmiuJl2XMBJnxKYeqRvLLrLS08s/edit?usp=sharing>). Corresponding or first-authors are welcome to submit the video. Please submit only one video per manuscript. The video can be emailed to contact@life-science-alliance.org

A. FINAL FILES:

B. MANUSCRIPT ORGANIZATION AND FORMATTING:

Sincerely,

Reviewer #1 (Comments to the Authors (Required)):

Summary of the paper:

Metastatic cancer cells migrate as clusters, referred here as spheroids, and have to face various physical constraints imposed by their environment. They obviously overcome these constraints as demonstrated by the lethal outcome of cancer metastasis. How do they succeed ? The authors address this issue from the original point of view of analysing the mechanical properties of human ovarian cancer cell spheroids. In this in vitro model, the cancer cells first form moruloid-like spheres that later change their morphology into blastuloid-like spheres. Blastuloids are characterized by the presence of a lumen and an ECM coat, both absent in moruloids. Using a microfluidic platform to spatially constrain the spheroids, the authors analyse the mechanical properties of the two types of spheroids and show that they display very different behaviours. In short, blastuloids are more resistant to physical constraints than moruloids. This resilience, as called by the authors, relies on the presence of their ECM, as shown by the authors in experiments whereby the ECM was eliminated, leading to lumen loss and greater disintegration of the blastoids, as predicted in computer simulations performed by the authors. Building on these observations, the authors analysed the behaviour of blastoids after the knock-down of an important molecule of the ECM, E-Cadherin, which is highly expressed in blastoids. Absence of E-Cadherin similarly led to a decrease formation of the lumen and greater disintegration of the blastoids. With this study, the authors provide good evidence for the relevance of the blastoid ECM to an elastic phenotype of blastoids facilitating their resistance (and hence cell survival) in spatially constrained spaces and flows.

Data:

Data provided in the first version and in this revised version convincingly support the following points:

- blastuloids have a higher elastic modulus than moruloids, travel faster through the microfluidic channels, undergo less disintegration and recover faster their shape after exiting the channels.
- the presence of a protein coat (ECM) on the spheroid is crucial for its resilience and dictates its mechanical behaviour.

I thank the authors for their additional work which has led to these new sets of data. I thank them as well for their very detailed answers and explanations to my questions. I am very satisfied and congratulate them for this work.

I only have a very tiny remark:

Figure 1Bi: the blastuloid is missing

Reviewer #2 (Comments to the Authors (Required)):

The authors have now performed extensive additional experiments, including KD studies E-cadherin and addressed my previous concerns adequately.

March 3, 2025

RE: Life Science Alliance Manuscript #LSA-2024-02601-TRR

Prof. Ramray Bhat
Indian Institute of Science Bangalore
Department of Developmental Biology and Genetics
GA07 New Biological Sciences Building
Bangalore, Karnataka 560012
India

Dear Dr. Bhat,

Thank you for submitting your Research Article entitled "Rheological transition driven by matrix makes cancer spheroids resilient under confinement". It is a pleasure to let you know that your manuscript is now accepted for publication in Life Science Alliance. Congratulations on this interesting work.

DISTRIBUTION OF MATERIALS:

Again, congratulations on a very nice paper. I hope you found the review process to be constructive and are pleased with how the manuscript was handled editorially. We look forward to future exciting submissions from your lab.

Sincerely,
